# Deep Learning Alternatives of the Kolmogorov Superposition Theorem

**Leonardo Ferreira Guilhoto**
Graduate Group in Applied Mathematics and Computational Science
University of Pennsylvania
Philadelphia, PA 19104, USA
`guilhoto@sas.upenn.edu`

**Paris Perdikaris**
Department of Mechanical Engineering & Applied Mechanics
University of Pennsylvania
Philadelphia, PA 19104, USA
`pgp@seas.upenn.edu`

## Abstract

This paper explores alternative formulations of the Kolmogorov Superposition Theorem (KST) as a foundation for neural network design. The original KST formulation, while mathematically elegant, presents practical challenges due to its limited insight into the structure of inner and outer functions and the large number of unknown variables it introduces. Kolmogorov-Arnold Networks (KANs) leverage KST for function approximation, but they have faced scrutiny due to mixed results compared to traditional multilayer perceptrons (MLPs) and practical limitations imposed by the original KST formulation. To address these issues, we introduce ActNet, a scalable deep learning model that builds on the KST and overcomes many of the drawbacks of Kolmogorov's original formulation. We evaluate ActNet in the context of Physics-Informed Neural Networks (PINNs), a framework well-suited for leveraging KST's strengths in low-dimensional function approximation, particularly for simulating partial differential equations (PDEs). In this challenging setting, where models must learn latent functions without direct measurements, ActNet consistently outperforms KANs across multiple benchmarks and is competitive against the current best MLP-based approaches. These results present ActNet as a promising new direction for KST-based deep learning applications, particularly in scientific computing and PDE simulation tasks.

## 1 Introduction

The Kolmogorov Superposition Theorem (KST) is a powerful and foundational result in mathematics, originally developed to solve Hilbert's 13th problem (Kolmogorov, 1961). Over time, it has gained significant attention as a tool for function approximation, showing that any multivariate continuous function can be represented exactly using finite sums and compositions of univariate functions. This theoretical insight has inspired numerous applications in mathematics and computational sciences since its inception in 1957, particularly in problems requiring efficient representation of complex functions in low-dimensional domains (Sprecher, 1996; Kůrková, 1992; Köppen, 2002).

Recently, Kolmogorov-Arnold Networks (KANs) (Liu et al., 2024) have renewed interest in the practical use of KST, especially in the context of physics-informed machine learning (PIML) (Karniadakis et al., 2021). Several papers have been published on this topic in the span of just a few months (Shukla et al., 2024; Wang et al., 2024c; Howard et al., 2024; Koenig et al., 2024; Rigas et al., 2024; Patra et al., 2024), demonstrating the current relevance of the KST for PIML research.

The KST's strength in approximating functions aligns well with problems in scientific computing and partial differential equation (PDE) simulation, where we often encounter functions representing

physical fields (e.g., velocity, temperature, pressure) that live in low-dimensional domains. Physics-Informed Neural Networks (PINNs) (Raissi et al., 2019; Wang et al., 2023) have emerged as a popular deep learning framework for such problems, offering a unique approach to learning latent functions without direct measurements by minimizing PDE residuals. By exploiting the KST, KANs offer a mathematically grounded approach to function approximation, sparking new research and development in the area of neural networks for numerically solving PDEs.

Despite its mathematical elegance, the original KST formulation is often impractical for real-world applications like PINNs, as the implementation of its univariate functions can be cumbersome Shukla et al. (2024); Yu et al. (2024). As a result, alternative formulations of the KST may be better equipped to address the challenges faced in modern approximation tasks, motivating ongoing research into more adaptable and scalable methods.

Alternative formulations of KST, which we will collectively refer to as **Superposition Theorems**, offer greater flexibility and more robust guarantees, making them better suited for modern deep learning applications, particularly scientific computing. These variants relax some of the original theorem's constraints, enabling more efficient computation and training of neural networks. Superposition Theorems open up new possibilities for designing deep learning architectures that can improve performance and scalability in challenging settings like PINNs.

The summary of contributions in this paper are as follows:

- We argue for using alternative versions of the KST for building neural network architectures, as shown in Table 1. Despite being the most popular iteration, Kolmogorov's original formula is the least suited for practical implementations.

- We propose ActNet, a novel neural network architecture leveraging the representation theorem from Laczkovich (2021) instead of the original KST employed in KANs. We prove ActNet's universal approximation properties using fixed depth and width, and propose a well-balanced initialization scheme, which assures that the activations of the network scale well with the size of the network, regardless of depth or width. Furthermore, ActNet does not have vanishing derivatives, making it ideal for physics-informed applications.

- We evaluate ActNet in the context of PINNs, demonstrating its potential in a setting that uniquely challenges current deep learning practices. Our physics-informed experiments showcase ActNet's ability to learn latent functions by minimizing PDE residuals, a task that aligns well with KST's strengths. ActNet outperforms KANs in every single experiment we examined, and appears to be competitive against the current best MLP-based approaches, including advanced methods such as PirateNets (Wang et al., 2024a).

Taken together, these contributions not only demonstrate the potential of ActNet as a novel architecture for tackling challenging problems in scientific computing and PDE simulation, but also open up a broader question: *which formulation of the KST is most suitable for deep learning applications?* While ActNet, leveraging Laczkovich's theorem (Laczkovich, 2021), shows promising results, it represents just a first step in exploring this rich space of possibilities. By addressing key limitations of existing KST-based approaches while maintaining their theoretical strengths, our work with ActNet aims to pave the way for future research into optimal KST formulations for neural networks.

## 2 SUPERPOSITION THEOREMS FOR REPRESENTING COMPLEX FUNCTIONS

Proposed in 1957 by Kolmogorov (1961), the Kolmogorov Superposition Theorem (KST, also known as Kolmogorov-Arnold representation theorem) offers a powerful theoretical framework for representing complex multivariate functions using sums and compositions of simpler functions of a single variable. This decomposition potentially simplifies the challenge of working with multivariate functions, breaking them down into manageable components.

In its original formulation, the KST states that any continuous function $f : [0, 1]^d \to \mathbb{R}$ can be represented *exactly* as

$$f(x_1, \ldots, x_d) = \sum_{q=0}^{2d} \Phi_q \left( \sum_{p=1}^{d} \phi_{q,p}(x_p) \right),$$

(1)

Table 1: Superposition formulas for a continuous function $f(x_1, \ldots, x_d)$. Kolmorogov's original result lead to many different versions of superpositions based on univariate functions. Although KANs employ the original formulation, other versions suggest distinct architectures. Other than the underlying formula, different theorems also use different assumptions about the domain of $f$ and regularity conditions of the inner/outer functions.

| Version | Formula | Inner Functions | Outer Functions | Other Parameters |
|---|---|---|---|---|
| Kolmogorov (1957) | $\sum_{q=0}^{2d} \Phi_q \left( \sum_{p=1}^{d} \phi_{q,p}(x_p) \right)$ | $2d^2$ | $2d$ | N/A |
| Lorentz (1962) | $\sum_{q=0}^{2d} g \left( \sum_{p=1}^{d} \lambda_p \phi_q(x_p) \right)$ | $2d$ | $1$ | $\lambda \in \mathbb{R}^d$ |
| Sprecher (1965) | $\sum_{q=0}^{2d} g_q \left( \sum_{p=1}^{d} \lambda_p \phi(x_p + qa) \right)$ | $1$ | $2d$ | $a \in \mathbb{R}, \ \lambda \in \mathbb{R}^d$ |
| Laczkovich (2021) | $\sum_{q=1}^{m} g \left( \sum_{p=1}^{d} \lambda_{pq} \phi_q(x_p) \right)$ | $m = \mathcal{O}(d)$ | $1$ | $\lambda \in \mathbb{R}^{d \times m}$ |

where $\Phi_q : \mathbb{R} \to \mathbb{R}$ and $\phi_{q,p} : [0,1] \to \mathbb{R}$ are univariate continuous functions known as *outer functions* and *inner functions*, respectively.

The first version of the KST was created to address Hilbert's 13th problem (Hilbert, 2000) on the representability of solutions to 7th-degree equations via continuous functions of at most two variables. Since its inception in the mid-20th century, the theorem has inspired numerous connections to approximation theory and neural networks, due to its potential for studying multivariate functions.

Over time, a multitude of different versions of KST have emerged, each with slightly different formulations and conditions, reflecting the evolving attempts to adapt the theorem to different applications and mathematical frameworks. Some of the most important variants are outlined in Table 1. For more information, we recommend the excellent book written by Sprecher (2017), who has worked on KST related problems for the past half century.

**KST And Neural Networks.** There is a substantial body of literature connecting the Kolmogorov Superposition Theorem (KST) to neural networks (Lin & Unbehauen, 1993; Köppen, 2002), notably pioneered by Hecht-Nielsen in the late 1980s (Hecht-Nielsen, 1987). KST was seen as promising for providing exact representations of functions, unlike approximation-based methods such as those of Cybenko (1989). More recently, Kolmogorov-Arnold Networks (KANs) (Liu et al., 2024) were proposed, attracting attention as a potential alternative to MLPs. Despite the initial enthusiasm, KANs have faced scrutiny due to mixed empirical performance and questions about experimental fairness (Yu et al., 2024; Shukla et al., 2024).

As detailed in Table 1, the KST exists in multiple versions, each with unique conditions and formulations, offering significant flexibility over how functions can be decomposed and represented. These KST variants provide a theoretical foundation for designing new neural network architectures, rooted in well-established function approximation theory. Despite serving as the basis for KANs, the original formulation of the KST in equation (1) offers the least information about the structure of the superposition formula. As can be seen in Table 1, Kolmogorov's formulation is in fact, the only one with $\mathcal{O}(d^2)$ unknown functions that have to be inferred, whereas other formulations scale linearly with the dimension $d$. This discrepancy motivates the use of other formulas from Table 1 in order to design neural network architectures based on superposition theorems.

**Kolmogorov's Theorem Can Be Useful, Despite Its Limitations.** Previous efforts to apply KST in function approximation have had mixed results (Sprecher, 1996; 2013; Köppen, 2002; Kůrková, 1992) as KST's non-constructive nature and irregularity of the inner functions (e.g., non-differentiability) can pose significant challenges. Vitushkin (1964) even proved that there exists certain analytical functions $f : [0,1]^d \to \mathbb{R}$ that are $r$ times continuously differentiable which cannot be represented by $r$ times continuously differentiable functions with less than $d$ variables, suggesting that in certain cases, the irregularity of inner functions is unavoidable.

The limitations of KST-based representations seem to be at direct odds with the many attempts over the past 50 years to use it as the basis for computational algorithms. The primary argument

against the use of KST for computation is the provable pathological behavior of the inner functions, as detailed in Vitushkin (1964). However, this impediment may not be directly relevant for most applications for a few different reasons.

1. **Approximation vs. Exact Representation**: The pathological behavior of inner functions mainly applies to exact representations, not approximations, which are more relevant in machine learning (Kůrková, 1992).

2. **Function-Specific Inner Functions**: Instead of using the same inner functions for all functions as suggested by most superposition theorems, allowing them to vary for specific target functions might address the irregularity problem in some cases.

3. **Deeper Compositions**: Deep learning allows for more layers of composition, which could help alleviate the non-smoothness in approximation tasks

While further research is needed to verify these claims, they offer practical insights that motivate continued exploration, including the work presented in this paper.

## 3 ACTNET - A KOLMOGOROV INSPIRED ARCHITECTURE

With renewed interest in KST and deep learning, many open questions remain about its applicability, limitations, and effective integration. In light of these questions, we propose **ActNet**, a deep learning architecture inspired by the Laczkovich (2021) version of the KST. This formulation implies several differences from KANs and other similar methods. The basic building block of an ActNet is an ActLayer, whose formulation is derived from the KST and can additionally be thought of as a multi-head MLP layer, where each head has a trainable activation function (hence the name _ActNet_).

### 3.1 THEORETICAL MOTIVATION

Theorem 1.2 in Laczkovich (2021) presents the following version of Kolmogorov's superposition theorem, which is the basis of the ActNet architecture:

**Theorem 3.1.** *Let $C(\mathbb{R}^d)$ denote the set of continuous functions from $\mathbb{R}^d \to \mathbb{R}$ and $m > (2 + \sqrt{2})(2d - 1)$ be an integer. There exists positive constants $\lambda_{ij} > 0$, $j = 1, \ldots, d; i = 1, \ldots, m$ and $m$ continuous increasing functions $\phi_i \in C(\mathbb{R})$, $i = 1, \ldots, m$ such that for every bounded function $f \in C(\mathbb{R}^d)$, there is a continuous function $g \in C(\mathbb{R})$ such that*

$$f(x_1, \ldots, x_d) = \sum_{i=1}^{m} g\left(\sum_{j=1}^{d} \lambda_{ij}\phi_i(x_j)\right).$$

*Using vector notation, this can equivalently be written as*

$$f(\boldsymbol{x}) = \sum_{i=1}^{m} g\left(\boldsymbol{\lambda}_i \cdot \phi_i(\boldsymbol{x})\right), \tag{2}$$

*where the $\phi_i$ are applied element-wise and $\boldsymbol{\lambda}_i = (\lambda_{i1}, \ldots, \lambda_{id}) \in \mathbb{R}^d$.*

As a corollary, we have that this expressive power is maintained if we allow the entries of $\lambda$ to be any number in $\mathbb{R}$, instead of strictly positive, and do not restrict the inner functions to be increasing[1]. Furthermore, if we define the matrices $\boldsymbol{\Phi}(x)_{ij} := \phi_i(x_j)$ and $\boldsymbol{\Lambda}_{ij} := \lambda_{ij}$ and let $S : \mathbb{R}^{a \times b} \to \mathbb{R}^a$ be the function that returns row sums of a matrix, or, if the input is a vector, the sum of its entries, then equation 2 can be equivalently expressed as

$$f(x) = \sum_{i=1}^{m} g\left(\boldsymbol{\lambda}_i \cdot \phi_i(\boldsymbol{x})\right) \tag{3}$$

$$= \sum_{i=1}^{m} g\left([S(\boldsymbol{\Lambda} \odot \boldsymbol{\Phi}(x))]_i\right) \tag{4}$$

$$= S\left(g\left[S(\boldsymbol{\Lambda} \odot \boldsymbol{\Phi}(x))\right]\right), \tag{5}$$

---

[1]An earlier version of ActNet attempted to impose these constraints to $\lambda$ and $\phi$, but this approach did not seem to lead to any practical gains, and potentially limited the expressive power of the network.

where $\odot$ is the Haddamard product between matrices.

From the formulations described in Table 1, we believe this one from Laczkovich (2021) is the best-suited for practical applications in deep learning. Some of the reasons for this are

- This version of the theorem is valid across all of $\mathbb{R}^d$, as opposed to being only applicable to the unit cube $[0,1]^d$, as is the case for most superposition theorems.
- It presents a flexible width size $m$, which yields exact representation as long as $m > (2+\sqrt{2})(2d-1)$, which scales linearly with the dimension $d$.
- It requires $\mathcal{O}(d)$ inner functions, as opposed to $\mathcal{O}(d^2)$ in the case of KANs.
- This formulation has other attractive interpretations in the light of recent deep learning advances, as detailed in section 3.4.

## 3.2 ACTNET FORMULATION

The main idea behind the ActNet architecture is implementing and generalizing the structure of the inner functions implied by theorem 3.1. That is, we create a computational unit called an ActLayer, which implements the forward pass $S(\boldsymbol{\Lambda} \odot \boldsymbol{\Phi}(\boldsymbol{x}))$, and then compose several of these units sequentially. This is done by parametrizing the inner functions $\phi_1, \ldots, \phi_m$ as linear compositions of a set of basis functions.

Given a set of univariate basis functions $b_1, \ldots, b_N : \mathbb{R} \to \mathbb{R}$, let $\boldsymbol{B} : \mathbb{R}^d \to \mathbb{R}^{N \times d}$ return the *basis expansion* matrix defined by $\boldsymbol{B}(x)_{ij} = b_i(x_j)$. The forward pass of the ActLayer component given an input $\boldsymbol{x} \in \mathbb{R}^d$ is then defined as

$$\text{ActLayer}_{\beta, \boldsymbol{\Lambda}}(\boldsymbol{x}) = S\left(\boldsymbol{\Lambda} \odot \beta \boldsymbol{B}(\boldsymbol{x})\right), \tag{6}$$

where the output is a vector in $\mathbb{R}^m$ and the trainable parameters are $\beta \in \mathbb{R}^{m \times N}$ and $\boldsymbol{\Lambda} \in \mathbb{R}^{m \times d}$. The graphical representation of this formula can be seen in Figure 1. Under this formulation, we say that the matrix $\boldsymbol{\Phi}(\boldsymbol{x}) = \beta \boldsymbol{B}(\boldsymbol{x}) \in \mathbb{R}^{m \times d}$ where $\boldsymbol{\Phi}(x)_{ij} = \phi_i(x_j)$ represent the *inner function expansion* of the layer and write

$$\boldsymbol{\Phi}(\boldsymbol{x}) = (\phi_1(\boldsymbol{x}), \ldots, \phi_m(\boldsymbol{x})) = \beta \boldsymbol{B}(\boldsymbol{x}), \tag{7}$$

where $\phi_k : \mathbb{R} \to \mathbb{R}$ are real-valued functions that are applied element-wise, taking the form

$$\phi_k(t) = \sum_{j=1}^{N} \beta_{kj} b_j(t). \tag{8}$$

Thus, the $k^{\text{th}}$ entry of an ActLayer output can be written as

$$\left(\text{ActLayer}_{\beta, \boldsymbol{\Lambda}}(\boldsymbol{x})\right)_k = \sum_{i=1}^{d} \lambda_{ki} \sum_{j=1}^{N} \beta_{kj} b_j(x_i) \tag{9}$$

$$= \sum_{i=1}^{d} \lambda_{ki} \phi_k(x_i) = \lambda_{k,:} \cdot \phi_k(\boldsymbol{x}), \tag{10}$$

where $\lambda_{k,:} = (\lambda_{k1}, \ldots, \lambda_{kd}) \in \mathbb{R}^d$ and $\phi_k(\boldsymbol{x}) = (\phi_k(x_1), \ldots, \phi_k(x_d)) \in \mathbb{R}^d$.

Additionally, before passing the input to ActLayers and before outputting the final result, ActNet uses linear projections from the latent dimension used for the ActLayers. We have empirically found this improved the performance of ActNet in some tasks by mixing the information of the different entries of $\boldsymbol{x}$. It is also possible to include an additive bias parameter at the end of the forward pass of each ActLayer. A visualization of the ActNet architecture can be seen in Figure 1.

Thus, the hyperparameters of an ActNet are: the embedding dimension $m \in \mathbb{N}$, which is used as the input and output dimension of the ActLayers; the number $N \in \mathbb{N}$ of basis functions used at each ActLayer; and the amount $L \in \mathbb{N}$ of ActLayers. Such a network has $\mathcal{O}(Lm(m+N))$ trainable parameters. For comparison, a KAN with $L$ layers of width $m$, spline order $K$ and grid definition $G$ has $\mathcal{O}(Lm^2(G+K))$ trainable parameters, which grows faster than ActNet. On the other hand,

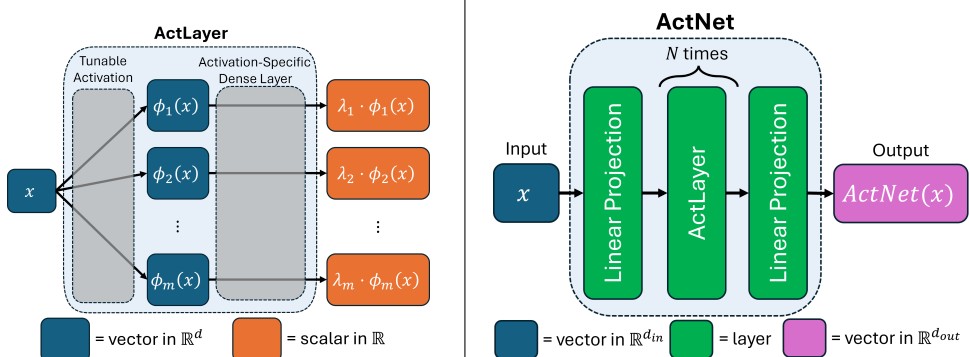

Figure 1: (left) Visual representation of an individual ActLayer. The ActLayer architecture can be seen as a MultiHead MLP layer with tunable activations. (right) Visual representation of the ActNet architecture. The input vector $x$ is first projected to an embedding dimension, then passed into $L$ composed blocks of ActLayer, and finally linearly projected into the desired output dimension.

an MLP of $L$ layers of width $m$ has $\mathcal{O}(Lm^2)$ trainable parameters. Our experiments indicate that $N$ can often be set as low as $N = 4$ basis functions, in which case the number of parameters of an ActNet is roughly similar to that of an MLP of the same width and depth, whereas KANs typically use $G > 100$, implying a much faster rate of growth. A comparison of the number of trainable parameters and flops between ActNet, KAN and MLPs can be seen in Appendix B, and Table 5.

Another advantage of the ActLayer over a KAN layer is that of simplicity in implementation. For example, using the `einsum` function found in packages like NumPy, PyTorch and JAX the forward pass of the ActLayer can even be implemented using a single line of code as `out=einsum('ij,jk,ik->k',`$B(\boldsymbol{x}),\beta,\lambda$`)`.

## 3.3 UNIVERSALITY

**Definition 3.2.** Given a set $\Omega \subseteq \mathbb{R}^d$, a neural network architecture is called a *universal approximator* on $\Omega$ if for every $f \in C(\Omega)$ and every $\varepsilon > 0$ there exists a choice of hyperparameters and trainable parameters $\theta$ such that the function $f_\theta$ computed by this neural network satisfies

$$\sup_{x \in \Omega} |f(x) - f_\theta(x)| < \varepsilon.$$

With this definition, we now state the universality of ActNet, with a proof presented in Appendix E.

**Theorem 3.3.** *A composition of two ActLayers is a universal approximator for any compact set $X \subseteq \mathbb{R}^d$. In other words, for any continuous function with compact domain $f : X \to \mathbb{R}$ and any $\varepsilon > 0$, there exists a composition of two ActLayers, where $|f(x) - Act(x)| < \varepsilon$ for all $x \in X$.*

In the construction implied by this proof, the first ActLayer has output dimension $m > (2 + \sqrt{2})(2d - 1)$, while the second ActLayer has output dimension 1. This means that we obtain universality using a width size that scales linearly with the input dimension $d$, potentially at the cost of requiring a large number $N$ of basis funcitons.

## 3.4 OTHER INTERPRETATIONS OF THE ACTLAYER

Although the representation formula from theorem 3.1 is the primary inspiration for the ActLayer, this unit can also be seen through other mathematically equivalent lenses. For example, the strategy of using several small units in parallel, followed by concatenating their outputs is exactly the idea behind multi-head attention employed in the transformer architecture from Vaswani (2017). In the same sense, **the ActLayer can be seen as a Multi-Head MLP Layer where each head has a different, tunable, activation function** $\phi_i$. In this analogy, each head first applies its individual activation function $\phi_i$, then applies a linear layer $\boldsymbol{\lambda}_i = \boldsymbol{\Lambda}_{i,:}$ with output dimension 1. Finally, the output of each head is concatenated together before being passed to the next layer.

Alternatively, the ActLayer can also be seen as a lighter version of a KAN layer, where each inner function $\phi_{pq}$ has the form $\phi_{pq}(x) = \lambda_{pq}\phi_q(x)$. Assuming all inner functions are parametrized by

the same set of basis functions, this is then equivalent to employing a KAN layer using Low-Rank Adaptation (LoRA) from Hu et al. (2021). This comparison however, overlooks the fact that the basis functions used for KAN are cubic B-splines, whereas ActNet employs a sinusoidal basis by default, as described in section 3.5.

All three interpretations are mathematically equivalent, and showcase potential benefits of the Act-Layer through different perspectives.

### 3.5 Choice of Basis functions and Initialization

Although ActNet can be implemented with any choice of basis functions, empirically we have observed robust performance using basis functions $b_1, \ldots, b_N : \mathbb{R} \to \mathbb{R}$ of the form

$$b_i(t) = \frac{\sin(\omega_i t + p_i) - \mu(\omega_i, p_i)}{\sigma(\omega_i, p_i)}, \tag{11}$$

where frequencies $w_i \sim N(0, 1)$ are initialized randomly at each layer from a standard normal distribution and phases $p_i$ are initialized at 0. We also add a small $\varepsilon > 0$ to the denominator in (11) for numerical stability when $\sigma(\omega_i, p_i)$ is small. In the equation above, the constants $\mu(\omega_i, p_i)$ and $\omega(\sigma_i, p_i)$ are defined as the mean and standard deviation, respectively, of the variable $Y = \sin(\omega_i X + p_i)$, when $X \sim N(0, 1)$. These values are expressed in closed form as

$$\mu(\omega_i, p_i) = \mathbb{E}[Y] = e^{\frac{-\omega^2}{2}} \sin(p_i), \tag{12}$$

$$\sigma(\omega_i, p_i) = \sqrt{\mathbb{E}[Y^2] - \mathbb{E}[Y]^2} = \sqrt{\frac{1}{2} - \frac{1}{2}e^{-2\omega_i^2}\cos(2p_i) - \mu(\omega_i, p_i)^2}. \tag{13}$$

This ensures that, if the inputs to the ActNet are normally distributed with mean 0 and variance 1 (a common assumption in deep learning), then the value of these basis functions will also have mean 0 and variance 1. After properly initializing the $\beta$ and $\lambda$ parameters (detailed in Appendix D.3), the central limit theorem then tells us that the output of an ActLayer will roughly follow a standard normal at initialization, which results in stable scaling of the depth and width of the network.

This can be formally stated as the theorem below, which implies that the activations of an ActNet will remain stable as depth and width increase. The proof can be seen in Appendix F.

**Theorem 3.4.** *At initialization, if the input $\boldsymbol{x} \in \mathbb{R}^d$ is distributed as $N(0, \boldsymbol{I}_d)$, then each entry of the output $ActLayer(\boldsymbol{x})$ has mean 0 and variance 1. In the limit as either the basis size $N \to \infty$ or the input dimension $d \to \infty$ we get that each output is distributed as a standard normal.*

## 4 Experiments

We believe that the framework of Physics Informed neural networks (PINNs) (Raissi et al., 2019) provides an ideal testbed for evaluating ActNet and other KST-inspired architectures. PINNs align well with the strengths of the Kolmogorov Superposition Theorem in several key aspects:

1. **Low-dimensional domains:** PINNs often deal with functions representing physical fields (e.g., velocity, temperature, pressure) that typically live in low-dimensional domains, where KST's function approximation capabilities have been better studied.

2. **Derivative information:** PINNs leverage derivative information through PDE residuals, a unique aspect that aligns with KST's ability to represent functions and their derivatives. In particular, as detailed in Appendix C, the derivative of an ActNet is another ActNet, which tells us the higher order derivatives of the network will not vanish, a frequent problem with ReLU MLPs and many of its variants.

3. **Complex functions:** PDEs often involve highly nonlinear and complex solutions, challenging traditional neural networks but potentially benefiting from KST-based approaches.

4. **Lack of direct supervision:** Unlike traditional supervised learning, PINNs often learn latent functions without direct measurements, instead relying on minimizing PDE residuals. This indirect learning scenario presents a unique challenge that has posed difficulties to existing architectures (Wang et al., 2024a) and may benefit from KST-inspired architectures.

Table 2: Best relative L2 errors and residual losses obtained by ablation experiments, as described in section 4.1. More details can be seen in Appendix G.

| Benchmark | Relative L2 Error (↓) | | | | Residual Loss (↓) | | | |
|---|---|---|---|---|---|---|---|---|
| | **ActNet** | **KAN** | **Siren** | **MLP** (base) | **ActNet** | **KAN** | **Siren** | **MLP** (base) |
| Poisson ($w = 16$) | **6.3e-4** | 1.8e-1 | 9.6e-2 | 6.6e-2 | **4.6e-3** | 2.6e0 | 8.9e-1 | 4.8e0 |
| Poisson ($w = 32$) | **6.3e-2** | 1.1e0 | 1.8e-1 | 1.0e0 | **8.0e-1** | 1.4e+3 | 8.4e+1 | 6.2e+4 |
| Helmholtz ($w = 16$) | **1.3e-3** | 2.8e-1 | 1.2e-1 | 4.6e-2 | **5.0e-3** | 2.6e0 | 9.2e-1 | 6.1e0 |
| Helmholtz ($w = 32$) | **1.1e-1** | 1.1e0 | 1.3e-1 | 1.0e0 | **5.7e-1** | 1.5e+3 | 8.1e+1 | 6.1e+4 |
| Allen-Cahn | 5.6e-5 | 5.3e-4 | **2.1e-5** | 1.1e-4 | **1.2e-8** | 3.8e-8 | 1.9e-8 | 2.0e-8 |

By focusing our experiments on PINNs, we not only aim to demonstrate ActNet's capabilities in a challenging and practically relevant setting, but also explore how KST-based approaches can contribute to scientific computing and PDE simulation. This choice of experimental framework allows us to evaluate ActNet's performance in scenarios that closely align with the theoretical strengths of the KST, potentially revealing insights that could extend to broader applications in the future.

Motivated by these observations, several papers (Shukla et al., 2024; Wang et al., 2024c; Howard et al., 2024; Shuai & Li, 2024; Koenig et al., 2024; Rigas et al., 2024; Patra et al., 2024) have been published attempting to specifically exploit KANs for physics informed training, with varying degrees of success. In the experiments that follow, we demonstrate that ActNet consistently outperforms KANs, both when using conducting our own experiments and when comparing against the published literature. Additionally, we also show that ActNet can be competitive against strong MLP baselines such as Siren (Sitzmann et al., 2020), as well as state-of-the-art architectures for PINNs such as the modified MLP from Wang et al. (2023) and PirateNets from Wang et al. (2024a).

## 4.1 ABLATIONS STUDIES

We compare the effectiveness of ActNet, KANs and MLPs in minimizing PDE residuals. In order to focus on PDE residual minimization, we enforce all initial and boundary conditions exactly, following Sukumar & Srivastava (2022), which removed the necessity for loss weight balancing schemes (Wang et al., 2022b), and thereby simplifies the comparisons. For all experiments, we train networks of varying parameter counts[2], and for each parameter size, with 12 different hyparaemeter configurations for each architecture. Additionally, for the sake of robustness, for each hyperparameter configuration we run experiments using 3 different seeds and take the median result. This means that for each PDE, 144 experiments were carried out per architecture type. We do this as our best attempt at providing a careful and thorough comparison between the methods.

Additionally, for the Poisson and Helmholtz PDEs, we consider 6 different forcing terms, with frequency parameters $w \in \{1, 2, 4, 8, 16, 32\}$. This means that for the Poisson PDE alone, 3,456 networks were trained independently, and similarly for Helmholtz. As the frequency parameter $w$ increases, solving the PDE becomes harder, as the model needs to adapt to very oscillatory PDE solutions. This is intended to showcase how ActNet's expressive power and training stability directly translate to performance in approximating highly oscillatory functions.

Table 2 summarizes the results of these ablations, where we find that ActNet outperforms KANs across all benchmarks, often by several orders of magnitude, and also outperforms both traditional MLPs as and strong MLP baselines like Siren (Sitzmann et al., 2020) on most cases.

**Poisson Equation.** We consider the 2D Poisson PDE on $[-1, 1]^2$ with Dirichlet boundary and forcing terms which yield exact solution $u(x, y) = \sin(\pi w x) \sin(\pi w y)$ for $w$ in $\{1, 2, 4, 8, 16, 32\}$, which are shown in Appendix G.3 as Figure 6, along with a full description of the PDE. ActNet consistently outperforms the other two types of architectures for all frequencies $w$. In particular, the difference in performance between methods becomes starker as the frequency $w$ increases. For the highest two frequencies of $w = 16$ and $w = 32$, ActNet can outperform KAN and Siren by as much as two orders of magnitude. The full results comparing ActNet, KANs, Siren and traditional MLPs can be seen on the table in Figure 8 of the Appendix.

---

[2]$\{10k, 20k, 40k, 80k\}$ for Poisson and Helmholtz PDEs, and $\{25k, 50k, 100k\}$ for Allen-Cahn.

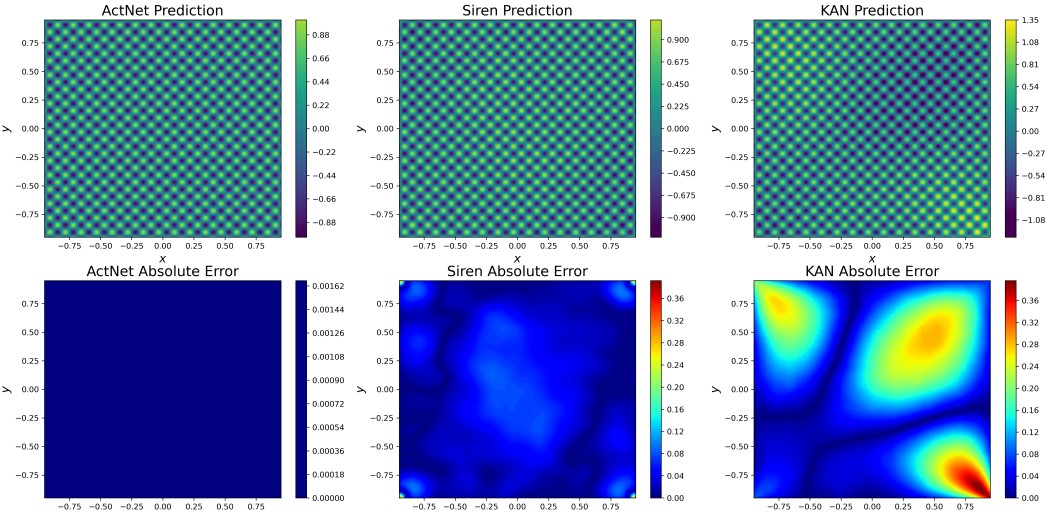

Figure 2: Example predictions for the Helmholtz equation using $w = 16$. The relative L2 errors for the ActNet, Siren and KAN solutions above are 1.04e-03, 8.82e-2 and 2.64e-1, respectively.

Table 3: Relative L2 errors of ActNet versus state-of-the-art results for PINNs.

| Benchmark | ActNet | JaxPi (Wang et al., 2023) | PirateNet (Wang et al., 2024a) |
|---|---|---|---|
| Advection ($c = 80$) | **9.50e-5** | 6.88e-4 | 5.48e-4 |
| Kuramoto–Sivashinsky (first time window) | 1.34e-5 | 1.42e-4 | **1.23e-5** |
| Kuramoto–Sivashinsky (full solution) | **8.53e-2** | 1.61e-1 | 2.11e-1 |

**Inhomogeneous Helmholtz Equation.** We consider the inhomogeneous Helmholtz PDE on $[-1, 1]^2$ with Dirichlet boundary condition, which is fully described in Appendix G.4. Similarly to what was done for the Poisson problem, we set forcing terms so this PDE has exact solution $u(x, y) = \sin(wx)\sin(wy)$, which can be seen in Figure 6 for different values of $w$ and $\kappa = 1$. Once again, ActNet outperforms both KANs and MLPs, across all frequencies (full details can be seen in Figure 10 in the Appendix), with a starker discrepancy for high values of $w$.

**Allen-Cahn Equation** On the Allen-Cahn PDE (fully described in Appendix G.5), ActNet outperforms KANs by around one order of magnitude across all network sizes. ActNet and Siren perform comparably, with ActNet yielding better results for smaller network sizes, and Siren performing better for larger network sizes, as can be seen in Figure 13. However, for larger networks, the relative L2 error is close to what can be achieved with single-precision computation for neural networks, so it is possible that the discrepancy arises due to floating-point error accumulation during training. Despite yielding slightly larger error in some cases, ActNet achieves the lowest final PDE residual loss across all network sizes compared to both Siren and KAN, as can be seen in Figure 14.

## 4.2 COMPARISONS AGAINST CURRENT STATE-OF-THE-ART

The goal of this section is to compare the performance of ActNet against some of the state-of-the-art results such as the modified MLP from the JaxPi library (Wang et al., 2023) and PirateNets (Wang et al., 2024a). As can be seen in Table 3, ActNet is capable of improving the best results available in the literature for two very challenging PINN problems.

**Advection Equation.** We consider the 1D advection equation with periodic boundary conditions, which has been extensively studied in Wang et al. (2023); Daw et al. (2022); Krishnapriyan et al. (2021) and is further described in Appendix G.6. Following Wang et al. (2023), we use initial conditions $g(x) = \sin(x)$ and the high transport velocity constant $c = 80$. This yields a challenging problem for PINNs, with highly oscillatory solution function. We compare ActNet against results

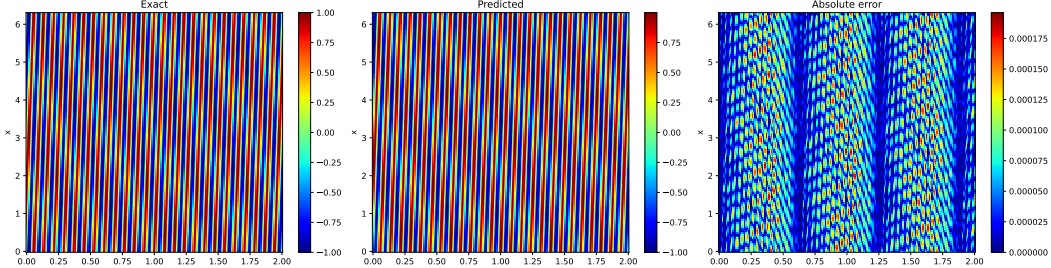

Figure 3: ActNet predictions for the advection equation ($c = 80$). The relative L2 error is 9.50e-5, whereas the best result found in the literature is 6.88e-4 (Wang et al., 2023).

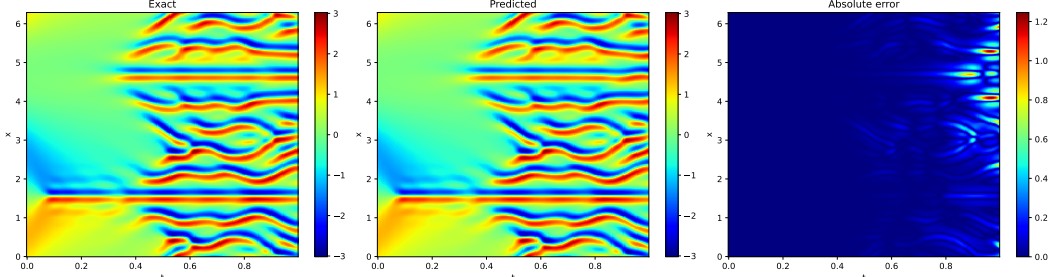

Figure 4: ActNet predictions for the chaotic Kuramoto–Sivashinsky PDE. The relative L2 error is 8.53e-2, whereas the best result found in the literature is 1.61e-1 (Wang et al., 2023).

from Wang et al. (2023), obtaining a relative L2 error of $9.50 \times 10^{-5}$, compared to their $6.88 \times 10^{-4}$ result, see Figure 3. On this same benchmark, PirateNet obtains a relative L2 error of $5.48 \times 10^{-4}$.

**Kuramoto–Sivashinsky Equation.** The Kuramoto–Sivashinsky equation (fully described in Appendix G.7) is a chaotic fourth-order non-linear PDE that models laminar flames. Due to its chaotic nature, slight inaccuracies in solutions quickly lead to diverging solutions. Training PINNs for this PDE usually requires splitting the time domain into smaller windows, sequentially solving one window at a time. Considering only the first window $t \in [0, 0.1]$, ActNet obtains a relative L2 error an order of magnitude lower than what is reported in Wang et al. (2023). For the full solution, ActNet also improves on the current best results from Wang et al. (2023) and PirateNet, see Figure 4.

## 5 CONCLUSION

**Summary.** We explore alternative formulations of the Kolmogorov Superposition Theorem (KST) to develop neural network architectures, and introduce ActNet. This architecture is inspired by Laczkovich's version of KST and is faster and simpler than Kolmogorov-Arnold Networks (KANs). We prove ActNet's universality for approximating multivariate functions and propose an initialization scheme to maintain stable activations. In physics-informed experiments, ActNet consistently outperforms KANs and is competitive with leading MLP architectures, even surpassing state-of-the-art models such as as PirateNet and the modified MLP of Wang et al. (2023) in some benchmarks.

**Limitations.** The primary limitation of ActNet and other KST-based approaches to deep learning is the slower computational speed compared to plain MLPs, although ActNet mitigates this to a considerable degree compared to KANs. Additionally, further optimizations may be possible, including hardware-level implementations tailored to ActNet's forward pass, similar to recent advances like FlashAttention for Transformers (Dao et al., 2022).

**Future Work.** Potential future directions in this field are determining the performance of ActNet and other KST-based architectures on data-driven problems, as well as studying the effect of replacing MLPs with these architectures inside larger neural networks such as transformers and U-Nets. We believe further study into KST-based architectures and specifically ActNet have the potential to advance not only scientific machine learning applications, but deep learning as a whole.

ETHICS STATEMENT

Our contributions enable advances in deep learning architectures. This has the potential to significantly impact a wide range of downstream applications. While we do not anticipate specific negative impacts from this work, as with any powerful predictive tool, there is potential for misuse. We encourage the research community to consider the ethical implications and potential dual-use scenarios when applying these technologies in sensitive domains.

REPRODUCIBILITY STATEMENT

Our entire code-base is publicly available online on GitHub. The code used to carry out the ablation experiments can be found at the GitHub repository `https://github.com/PredictiveIntelligenceLab/ActNet`. Our ActNet implementation is also available in the open-source JaxPi framework at `https://github.com/PredictiveIntelligenceLab/jaxpi/tree/ActNet`, which we used to carry out the comparisons against current state-of-the-art.

AUTHOR CONTRIBUTIONS

L.F.G. conceived the methodology and conducted the experiments. P.P. provided funding and supervised this study. All authors helped conceptualize experiments and reviewed the manuscript.

ACKNOWLEDGMENTS

We would like to acknowledge support from the US Department of Energy under the Advanced Scientific Computing Research program (grant DE-SC0024563) and the US National Science Foundation (NSF) Soft AE Research Traineeship (NRT) Program (NSF grant 2152205). We also thank Maryl Harris for helpful feedback when reviewing the writing of the manuscript and the developers of software that enabled this research, including JAX (Bradbury et al., 2018), Flax (Heek et al., 2023) Matplotlib (Hunter, 2007) and NumPy (Harris et al., 2020).

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

# Appendix

## A  MATHEMATICAL NOTATION

Table 4 summarizes the symbols and notation used in this work.

Table 4: Summary of the symbols and notation used in this paper.

| Symbol | Meaning |
|---|---|
| $\boldsymbol{x}$ | A vector $\boldsymbol{x} = (x_1, \dots, x_d)$ in $\mathbb{R}^d$ |
| $C(\Omega)$ | The set of continuous functions from a set $\Omega$ to $\mathbb{R}$ |
| $g$ | Outer function implied by a superposition theorem (see table 1) |
| $\phi_i$ | Inner functions implied by a superposition theorem (see table 1) |
| $\boldsymbol{\Phi}(\boldsymbol{x})$ | Inner function expansion matrix with $ij$ entry equal to $\phi_i(x_j) \in \mathbb{R}$ |
| $b_i$ | Basis function from $\mathbb{R}$ to $\mathbb{R}$ |
| $\boldsymbol{B}(\boldsymbol{x})$ | Basis expansion matrix with $ij$ entry equal to $b_i(x_j) \in \mathbb{R}$ |
| $\boldsymbol{\Lambda}$ | Matrix with $ij$ entry equal to $\lambda_{ij} \in \mathbb{R}$ |
| $S$ | Function that returns row sums of a matrix, or the sum of entries of a vector |
| $\odot$ | Hadamard (element-wise) product between vectors/matrices |
| $\theta$ | Parameters of a neural network model |
| $\boldsymbol{I}_d$ | The identity matrix of size $d$ |
| $\mathbb{E}$ | Expectation operand |

## B  COMPARISON ON PARAMETERS & FLOPS

From equation (9), each of the $k = 1, \dots m$ outputs of an ActLayer with input dimension $d$, output dimension $m$ and $N$ basis functions is defined as

$$\left(\text{ActLayer}_{\beta, \boldsymbol{\Lambda}}(\boldsymbol{x})\right)_k = \sum_{i=1}^{d} \sum_{j=1}^{N} \lambda_{ki} \beta_{kj} b_j(x_i).$$

This sum entails $\mathcal{O}(dN)$ operations per output. Since there are $m$ outputs total, the computational complexity of this ActLayer is $\mathcal{O}(mdN)$. This means that this algorithmic complexity is a little slower than that of the traditional dense layer of an MLP (which is $\mathcal{O}(dm)$), but around the same order of magnitude for low values of $N$. **In our experiments, we have observed that the basis size $N$ usually does not need to be large, with $N = 4$ usually yielding the best results. In particular, in all experiments where ActNet beats state-of-the-art results, we do so by setting $N = 4$ basis function**, as can be seen in section 4.2. This setting makes an ActNet around $\approx$2-3 times slower than an MLP of the same size. To see this phenomena in practical terms, we direct the reader to table 9, where computational times for the Allen-Cahn PDE are compared for setting $N$ in $\{4, 8, 16\}$.

In the case of a KAN layer, the algorithmic complexity is $\mathcal{O}(mdK(G + K))$ Yu et al. (2024), where $G$ is the grid size hyper-parameter, and $K$ is the spline order (usually taken to be $K = 3$). Unlike ActNet, however, typical values of $G$ can range from around 5 to several hundred, with the original KAN paper (Liu et al., 2024) using experiments with as high as $G = 1000$, which makes the architecture impractical for realistic values of widths and depths. Once again, to see this phenomena in practical terms, we direct the reader to table 9, where we set $G = N$ in $\{4, 8, 16\}$. This comparison showcases another advantage of ActNets over KANs.

A summary of the hyperparameters of ActNet, KAN and MLP, along with their implied parameter count and flops of a forward pass can be seen in table 5.

Table 5: Table comparing parameter and algorithmic complexity of ActNet, KAN and MLP. Since the typical value of $N$ is low, ActNet performs at slower, but comparable speed to MLPs, whereas KANs become very slow as the value of $G$ increases.

| Architecture | Hyperparameters | Parameter Count | FLOPs |
|---|---|---|---|
| ActNet | depth $L$
hidden dim. $m$
basis size $N$ | $\mathcal{O}(Lm(m+N))$ | $\mathcal{O}(Lm^2N)$ |
| KAN | depth $L$
hidden dim. $m$
grid size $G$
spline order $K$ | $\mathcal{O}(Lm^2(G+K))$ | $\mathcal{O}(Lm^2K(G+K))$ |
| MLP | depth $L$
hidden dim. $m$ | $\mathcal{O}(Lm^2)$ | $\mathcal{O}(Lm^2)$ |

## C  THE DERIVATIVE OF AN ACTNET IS ANOTHER ACTNET

From equation (9), we get that the partial derivative of the $k$th output of an ActLayer with respect to $x_l$ is

$$\frac{\partial}{\partial x_l}\left(\text{ActLayer}_{\beta,\Lambda}(\boldsymbol{x})\right)_k = \frac{\partial}{\partial x_l}\left(\sum_{i=1}^{d}\lambda_{ki}\sum_{j=1}^{N}\beta_{kj}b_j(x_i)\right)$$

$$= \lambda_{kl}\sum_{j=1}^{N}\beta_{kj}\frac{\partial}{\partial x_l}b_j(x_l)$$

$$= \lambda_{kl}\sum_{j=1}^{N}\beta_{kj}b_j'(x_l)$$

$$= \lambda_{kl}\phi_k'(x_l),$$

where $\phi_k'(t) = \sum_{j=1}^{N}\beta_{kj}b_j'(x_l)$ is the derivative of the $k$th inner function $\phi_k(t) = \sum_{j=1}^{N}\beta_{kj}b_j(x_l)$.

This formula then tells us that the Jacobian $J_{\text{ActLayer}_{\beta,\Lambda}}(\boldsymbol{x})$ of an ActLayer is

$$J_{\text{ActLayer}_{\beta,\Lambda}}(\boldsymbol{x}) = \Lambda \odot \Phi'(\boldsymbol{x}), \tag{14}$$

where $\Phi'(\boldsymbol{x}) \in \mathbb{R}^{m \times d}$ is the matrix defined by $\Phi'(\boldsymbol{x})_{ij} = \phi_i'(x_j)$. This formulation is precisely what we compute in the forward pass $\text{ActLayer}_{\beta,\Lambda}(\boldsymbol{x}) = S(\Lambda \odot \Phi(\boldsymbol{x}))$, with the caveat that this Jacobian is now a matrix instead of a single vector.

If the basis functions $b_1, \ldots, b_N$ are picked so that their derivatives do not deteriorate, as is the case with the sinusoidal basis proposed in section 3.5, then the universality result of theorem 3.3 tells us that a sufficiently large ActNet will be able to approximate derivatives up to any desired precision $\varepsilon > 0$. This statement once again makes the case for the potential of ActNet for Physics Informed Machine Learning, where a network needs to learn to approximate derivatives of a function, instead of point values.

## D  ACTNET IMPLEMENTATION DETAILS

Our implementation of ActNet is carried out in Python using JAX (Bradbury et al., 2018) and Flax (Heek et al., 2023), although the architecture is also easily compatible with other deep learning frameworks such as PyTorch (Ansel et al., 2024) and TensorFlow Abadi et al. (2015). The code is available at https://github.com/PredictiveIntelligenceLab/ActNet.

### D.1  FREQUENCY PARAMETER $\omega_0$

In a manner analogous to what is done in Siren (Sitzmann et al., 2020) and random Fourier features (Tancik et al., 2020), we have observed that ActNet benefits from having a $\omega_0 > 0$ parameter that

Table 6: Initialization schemes for an ActLayer with input dimension $d$, output dimension $m$ and $N$ basis functions. Each entry of a parameter is sampled in an *i.i.d.* manner.

| Trainable Parameter | Dimension | Mean of Entries | Std. Dev. of Entries | Distribution |
|---|---|---|---|---|
| $\beta$ | $\mathbb{R}^{m \times N}$ | 0 | $\frac{1}{\sqrt{N}}$ | Gaussian/uniform |
| $\mathbf{\Lambda}$ | $\mathbb{R}^{m \times d}$ | 0 | $\frac{1}{\sqrt{d}}$ | Gaussian/uniform |
| $\omega$ | $\mathbb{R}^{N}$ | 0 | 1 | Gaussian |
| $p$ | $\mathbb{R}^{N}$ | 0 | 0 | constant |
| bias (optional) | $\mathbb{R}^{m}$ | 0 | 0 | constant |

multiplies the original input to the network. Using such a parameter better allows the architecture to create solutions that attain to a specific frequency content, using higher values of $\omega_0$ for approximating highly oscillatory functions and smaller values for smoother ones. This parameter can be trainable, but we generally found it best to set it to a fixed value, which may be problem dependent, much like the case with Siren and Fourier Features.

## D.2 TRAINING FREQUENCIES AND PHASES

Additionally, it is possible to allow the frequencies $\omega_i$ and phases $p_i$ of that basis functions $b_i$ from equation (11) to be trainable parameters themselves, making the basis functions at each layer customizable to the problem at hand. However, the parameters $\omega_i$ and $p_i$ of the basis functions appear to be very sensitive during training, and optimization via stochastic gradient-based methods sometimes lead to unstable behavior. This issue, however, can be mitigated by using Adaptive Gradient Clipping (AGC) proposed in Brock et al. (2021), where each parameter unit has a different clip depending on the magnitude of the parameter vector. We empirically observe that a hyperparameter of $1e - 2$ works well for training basis functions of an ActNet.

## D.3 INITIALIZATION SCHEME

The parameters of an ActLayer are initialized according to table 6. As will be shown later in the proof of theorem 3.4, using these initializations and applying the central limit theorem then tells us that the output of an ActLayer, as defined in (6), will be normally distributed with mean 0 and variance 1, as long as either $N$ or $d$ are sufficiently large. This same argument can then be applied inductively on the composed layers of the ActNet to get a statement on the stability of the statistics of activations throughout the entire network.

The $\beta$ and $\mathbf{\Lambda}$ parameters can be initialized using either a Gaussian or uniform distribution. Having said that, for the sake of consistency we use uniform distributions for initializing these parameters for all experiments in section 4.

## D.4 CHOICE OF HYPERPARAMETERS

Results from the ablation studies from section 4 can also be used to gain information over reasonable hyperparameter choices for the ActNet architecture. A representative plot of the choice of hyperparameters for the Allen-Cahn PDE can be seen in figure 5. Overall, our the main "rule-of-thumb" heuristics we have learned are:

- The basis size $N$ does not need to be large for accurate results. In fact, except for small parameter counts and shallow depths, we don't see accuracy gains for increasing the value of $N$ beyond 4 or so. At the same time, as detailed in appendix B, the computational complexity of the model increases for larger $N$ (even when network size is fixed). Therefore, we recommend first setting the basis size to around $N = 4$, and increasing this value only if needed.
- As is often the case for deep learning, we find that composing more layers and increasing network width generally improves performance by increasing network capacity. Under

fixed network size, choosing to increase depth implies a decrease in width, and vice versa. While choosing the right combination of depth/width is likely problem dependent (as is the case with MLPs), we recommend setting the number of ActLayers in an ActNet at least $\geq 2$. Not only do we empirically observe a large performance boost over using a single ActLayer, this is also the minimum required depth for the theoretical guarantee of universal approximation, as described by theorem 3.3.

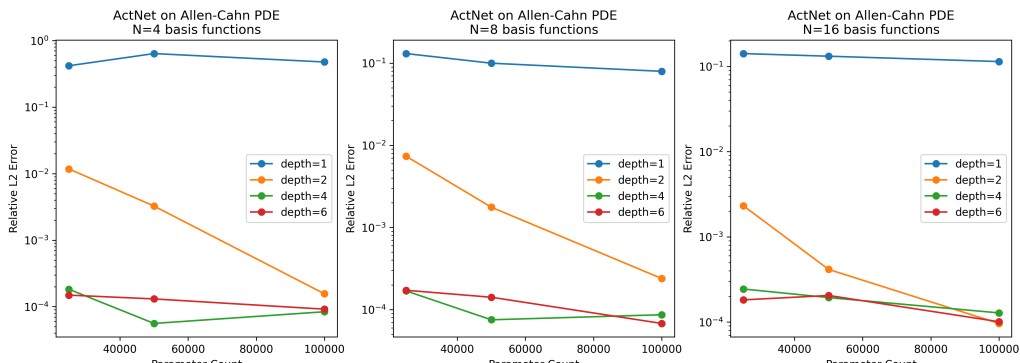

Figure 5: ActNet performance (relative L2 error) on the Allen-Cahn PDE under different hyperparameter settings. After selecting network depth $L$ and number of basis functions $N$, the width $m$ of networks was computed in order to satisfy the required parameter size. As such, for a given network size, larger values of deph imply smaller widths, and vice-versa. The values plotted for each hyperparameter configuration is the median from 3 runs using different seeds.

## E    PROOF OF THEOREM 3.3

*Proof of theorem 3.3.* By KST (theorem 3.1), we know that for any $m > (2 + \sqrt{2})(2d - 1)$, the function $f : \mathbb{R}^d \to \mathbb{R}$ has a representation of the form

$$f(x) = \sum_{q=1}^{m} g\left(\sum_{p=1}^{d} \lambda_{pq}\phi_q(x_p)\right),$$

where $g : \mathbb{R} \to \mathbb{R}$ and $\phi_1, \ldots, \phi_m : \mathbb{R} \to \mathbb{R}$ are continuous univariate functions and $\lambda_{pq} > 0$ are positive scalars.

Now, let $Y := \left\{\sum_{p=1}^{d} \lambda_{pq}\phi_q(x_p) | x \in X, q = 1, \ldots, m\right\}$ be the set of possible inputs to $g$. Since $X$ is compact and each of the $\phi_q$ is continuous, this means that $Y$ must be compact as well.

Now let $\varepsilon' := \varepsilon/m$. Since $g$ is continuous and its inputs are confined to a compact domain $Y$, it must be uniformly continuous. That is, there exists some $\delta > 0$ such that for all $y, z \in Y$, if $|z - y| < \delta$ then $|g(z) - g(y)| < \varepsilon'/2$. Additionally, due to Weierstrass theorem[3] we know that there exists a polynomial $poly_g : \mathbb{R} \to \mathbb{R}$ of degree $N_g$ such that for any $y \in Y$ we have that $|g(y) - poly_g(y)| < \varepsilon'/2$. Therefore, we get that

$$\begin{aligned}|g(z) - poly(y)| &= |g(z) - g(y) + g(y) - poly_g(y)| \\ &\leq |g(z) - g(y)| + |g(y) - poly_g(y)| \\ &< \varepsilon'/2 + \varepsilon'/2 = \varepsilon'.\end{aligned}$$

---

[3]Weierstrass theorem states that polynomials are dense in $C(X)$ for any compact $X \subset \mathbb{R}$.

Which implies that if $z_1, \ldots, z_m$ and $y_1, \ldots, y_m$ are scalars such that $|z_q - y_q| < \delta$ for all $q = 1, \ldots, m$, then we have

$$\left| \sum_{q=1}^{m} g(z_q) - \sum_{q=1}^{m} poly_g(y_q) \right| \leq \sum_{q=1}^{m} |g(z_q) - poly_g(y_q)|$$

$$< \sum_{q=1}^{m} \varepsilon' = m\varepsilon' = \varepsilon.$$

Therefore, if we can approximate the inner functions $\phi_q(t)$ using polynomials $poly_q(t)$ and make sure that $\sum_{p=1}^{d} \lambda_{pq} \phi_q(x_p)$ is at most $\delta$ far from the approximation $\sum_{p=1}^{d} \lambda_{pq} poly_q(x_p)$ for all $q = 1, \ldots, m$, we can finish the proof of universality for ActNet.

To prove this final statement, we first set $M := \max\{\lambda_{pq} | p = 1, \ldots, d; \ q = 1, \ldots, m\}$ and then note that by Weierstrass' Theorem, for each $\phi_q$ there exists a polynomial $poly_q$ of degree $N_q$ such that $|\phi_q(z) - poly_q(z)| < \frac{\delta}{dM}$ for all $z \in Y$. Thus, we have that

$$\left| \sum_{p=1}^{d} \lambda_{pq} \phi_q(x_p) - \sum_{p=1}^{d} \lambda_{pq} poly_q(x_p) \right| \leq \sum_{p=1}^{d} \lambda_{pq} |\phi_q(x_p) - poly_q(x_p)|$$

$$< \sum_{p=1}^{d} \lambda_{pq} \frac{\delta}{dM}$$

$$\leq \sum_{p=1}^{d} \frac{\delta}{d}$$

$$= \delta,$$

which completes the proof. $\square$

## F   Proof of Statistical Stability of Activations Through Layers

Before proving theorem 3.4, we prove the following lemma, where we compute the formulas stated in equations (12-13)

**Lemma F.1.** *Let $\omega, p \in \mathbb{R}$ be fixed real numbers and $X \sim N(0,1)$ be a random variable distributed as a standard normal. Then the random variable $Y = \sin(\omega X + p)$ has mean and variance as follows:*

$$\mathbb{E}[Y] = e^{\frac{-\omega^2}{2}} \sin(p), \tag{15}$$

$$Var[Y] = \frac{1}{2} - \frac{1}{2} e^{-2\omega^2} \cos(2p) - e^{-\omega^2} \sin(p)^2. \tag{16}$$

*Proof.* First, we note that for any $a > 0$ and $b \in \mathbb{C}$ we have that

$$\int_{-\infty}^{\infty} e^{-ax^2 + bx} = \frac{\pi}{a} e^{\frac{b^2}{4a}}. \tag{17}$$

Now, using this fact and the complex representation $\sin(\theta) = \frac{e^{i\theta} - e^{-i\theta}}{2i}$ we have that

$$
\begin{aligned}
\mathbb{E}[Y] &= \int_{-\infty}^{\infty} \sin(\omega x + p) \left( \frac{1}{\sqrt{2\pi}} e^{\frac{-x^2}{2}} \right) dx \\
&= \frac{1}{\sqrt{2\pi}} \int_{-\infty}^{\infty} \frac{e^{i(\omega x + p)} - e^{-i(\omega x + p)}}{2i} e^{\frac{-x^2}{2}} dx \\
&= \frac{1}{2i\sqrt{2\pi}} \int_{-\infty}^{\infty} \left( e^{i(\omega x + p) - \frac{x^2}{2}} - e^{-i(\omega x + p) - \frac{x^2}{2}} \right) dx \\
&= \frac{1}{2i\sqrt{2\pi}} \left( \int_{-\infty}^{\infty} e^{-\frac{1}{2}x^2 + i\omega x + ip} dx - \int_{-\infty}^{\infty} e^{-\frac{1}{2}x^2 - i\omega x - ip} dx \right) \\
&= \frac{1}{2i\sqrt{2\pi}} \left( e^{ip} \int_{-\infty}^{\infty} e^{-\frac{1}{2}x^2 + i\omega x} dx - e^{-ip} \int_{-\infty}^{\infty} e^{-\frac{1}{2}x^2 - i\omega x} dx \right) \\
&= \frac{1}{2i\sqrt{2\pi}} \left( e^{ip} e^{\frac{-w^2}{2}} \sqrt{2\pi} - e^{-ip} e^{-\frac{w^2}{2}} \sqrt{2\pi} \right) \\
&= \frac{1}{2i} \left( e^{ip} e^{\frac{-w^2}{2}} - e^{-ip} e^{-\frac{w^2}{2}} \right) \\
&= e^{-\frac{w^2}{2}} \frac{e^{ip} - e^{-ip}}{2i} \\
&= e^{-\frac{w^2}{2}} \sin(p),
\end{aligned}
$$

which proves the claim for the expectation of $Y$. As for the variance, we first compute

$$
\begin{aligned}
\mathbb{E}[Y^2] &= \int_{-\infty}^{\infty} \sin(\omega x + p)^2 \left( \frac{1}{\sqrt{2\pi}} e^{\frac{-x^2}{2}} \right) dx \\
&= \frac{1}{\sqrt{2\pi}} \int_{\infty}^{\infty} \frac{e^{2i(\omega x + p)} + e^{-2i(\omega x + p)} - 2}{-4} e^{\frac{-x^2}{2}} dx \\
&= -\frac{1}{4\sqrt{2\pi}} \int_{-\infty}^{\infty} \left( e^{2i(\omega x + p)} e^{\frac{-x^2}{2}} + e^{-2i(\omega x + p)} e^{\frac{-x^2}{2}} - 2e^{\frac{-x^2}{2}} \right) dx \\
&= -\frac{1}{4\sqrt{2\pi}} \left( \int_{-\infty}^{\infty} e^{2i(\omega x + p)} e^{\frac{-x^2}{2}} dx + \int_{-\infty}^{\infty} e^{-2i(\omega x + p)} e^{\frac{-x^2}{2}} dx - \int_{-\infty}^{\infty} 2e^{\frac{-x^2}{2}} dx \right) \\
&= -\frac{1}{4\sqrt{2\pi}} \left( \int_{-\infty}^{\infty} e^{-\frac{1}{2}x^2 + 2i\omega x + 2ip} dx + \int_{-\infty}^{\infty} e^{-\frac{1}{2}x^2 - 2i\omega x - 2ip} dx - 2\sqrt{2\pi} \right) \\
&= -\frac{1}{4\sqrt{2\pi}} \left( e^{2ip} \int_{-\infty}^{\infty} e^{-\frac{1}{2}x^2 + 2i\omega x} dx + e^{-2ip} \int_{-\infty}^{\infty} e^{-\frac{1}{2}x^2 - 2i\omega x} dx \right) + \frac{1}{2} \\
&= -\frac{1}{4\sqrt{2\pi}} \left( e^{2ip} e^{\frac{-4\omega^2}{2}} \sqrt{2\pi} + e^{-2ip} e^{\frac{-4\omega^2}{2}} \sqrt{2\pi} \right) + \frac{1}{2} \\
&= -e^{-2\omega^2} \left( \frac{e^{2ip} + e^{-2ip}}{4} \right) + \frac{1}{2} \\
&= -e^{-2\omega^2} \frac{\cos(2p)}{2} + \frac{1}{2} \\
\\
&= \frac{1}{2} - \frac{1}{2} e^{-2\omega^2} \cos(2p),
\end{aligned}
$$

which means that

$$
\begin{aligned}
Var[Y] &= \mathbb{E}[Y^2] - \mathbb{E}[Y]^2 \\
&= \frac{1}{2} - \frac{1}{2} e^{-2\omega^2} \cos(2p) - e^{-w^2} \sin(p)^2,
\end{aligned}
$$

thus completing the proof. $\qquad \square$

We are now ready to prove the main result of this section.

*Proof of theorem 3.4.* The $k$th output of an ActLayer is computed as

$$\left(\text{ActLayer}_{\beta,\boldsymbol{\Lambda}}(\boldsymbol{x})\right)_k = \sum_{i=1}^{d} \sum_{j=1}^{N} \lambda_{ki} \beta_{kj} b_j(x_i).$$

Therefore, by linearity of expectation and then lemma F.1 we have

$$\mathbb{E}_{\boldsymbol{x}\sim N(0,I)} \left[\left(\text{ActLayer}_{\beta,\boldsymbol{\Lambda}}(\boldsymbol{x})\right)_k\right] = \sum_{i=1}^{d} \sum_{j=1}^{N} \lambda_{ki} \beta_{kj} \mathbb{E}_{\boldsymbol{x}\sim N(0,I)} \left[b_j(x_i)\right]$$

$$= \sum_{i=1}^{d} \sum_{j=1}^{N} \lambda_{ki} \beta_{kj}.$$

Since $\boldsymbol{\Lambda}$ and $\beta$ are independent and all entries have mean 0, this means that the $k$th output has mean 0. As for the variance $\sigma_k^2$, we now compute

$$\sigma_k^2 = \mathbb{E}\left[\left(\text{ActLayer}_{\beta,\boldsymbol{\Lambda}}(\boldsymbol{x})\right)_k^2\right] \tag{18}$$

$$= \mathbb{E}\left[\left(\sum_{i=1}^{d} \sum_{j=1}^{N} \lambda_{ki} \beta_{kj} b_j(x_i)\right)^2\right] \tag{19}$$

$$= \mathbb{E}\left[\left(\sum_{i=1}^{d} \sum_{j=1}^{N} \sum_{a=1}^{d} \sum_{b=1}^{N} \lambda_{ki} \beta_{kj} b_j(x_i) \lambda_{ka} \beta_{kb} b_b(x_a)\right)\right] \tag{20}$$

$$= \sum_{i=1}^{d} \sum_{j=1}^{N} \sum_{a=1}^{d} \sum_{b=1}^{N} \mathbb{E}\left[\lambda_{ki} \beta_{kj} b_j(x_i) \lambda_{ka} \beta_{kb} b_b(x_a)\right] \tag{21}$$

$$= \sum_{i=1}^{d} \sum_{j=1}^{N} \mathbb{E}\left[\lambda_{ki}^2 \beta_{kj}^2 b_j(x_i)^2\right] \tag{22}$$

$$= \sum_{i=1}^{d} \sum_{j=1}^{N} \mathbb{E}\left[\lambda_{ki}^2\right] \cdot \mathbb{E}\left[\beta_{kj}^2\right] \cdot \mathbb{E}_{\boldsymbol{x}\sim N(0,I)} \left[b_j(x_i)^2\right] \tag{23}$$

$$= \sum_{i=1}^{d} \sum_{j=1}^{N} \frac{1}{d} \cdot \frac{1}{N} \cdot 1 \tag{24}$$

$$= 1. \tag{25}$$

In the deduction above, to go from line (21) to line (22) we use the fact that the expectation of products of independent variables is the product of expectations, then recall that $\mathbb{E}\left[b_j(x_i)\right] = 0$, which allows us to cancel the terms where $i \neq a$ and $j \neq b$.

This proves the first part of the theorem (outputs have mean 0 and variance 1 at initialization). Now to prove the second part (outputs are distributed normally as either $N$ or $d$ go to infinity), we note that from the deduction above, for any fixed $i = 1, \ldots, d$, the random variable $X_i^d \in \mathbb{R}$ defined by

$$X_i^d := \sum_{j=1}^{N} \lambda_{ki} \beta_{kj} b_j(x_i),$$

has mean 0 and variance $\frac{1}{d}$.

Therefore, by the central limit theorem, we have that

$$\left(\text{ActLayer}_{\beta,\boldsymbol{\Lambda}}(\boldsymbol{x})\right)_k = \sum_{i=1}^{d} X_i^d,$$

will be distributed as a standard normal $N(0,1)$ as $d \to \infty$. An analogous argument can be made, swapping the role of $d$ and $N$, which finishes the proof for the theorem. $\qquad\square$

## G   EXPERIMENTAL DETAILS

We dedicate the remaining of the appendix for detailing the experimental setup used to produce the results indicated in section 4. As a general rule, we used large batch sizes whenever possible, as this has been shown to improve training in PINNs Sankaran et al. (2022), and enforced boundary conditions exactly when possible, following the strategy in Sukumar & Srivastava (2022).

We would also like to highlight that while we focus our experiments on the data-free paradigm of Physics Informed Neural Networks, other methods exist for learning solutions of PDEs from data (Gin et al., 2021; Lusch et al., 2018). Notably, methods in the field of operator learning such as (Li et al., 2021a; Lu et al., 2021; Kissas et al., 2022; Li et al., 2020; Wang et al., 2022a; Guilhoto & Perdikaris, 2024) allow for predicting entire families of PDEs using a single model, and may even be trained with the aid of a physics informed loss (Li et al., 2021b; Wang et al., 2021). While we used PINNs as a first testbed for ActNet, we believe the architecture has potential application to data-driven methods as well, and leave further exploration as future research.

### G.1   KAN IMPLEMENTATION DETAILS

All experiments were conducted in Python using JAX (Bradbury et al., 2018). Since the original implementation of KANs is not GPU-enabled and therefore very slow, we instead reference the "Efficient KAN" code found in `https://github.com/Blealtan/efficient-kan.git`. Since Efficient KANs were implemented originally in PyTorch (Ansel et al., 2024), we translated the code to Flax (Heek et al., 2023). This implementation can be found at `https://github.com/PredictiveIntelligenceLab/ActNet`.

### G.2   HYPERPARAMETER ABLATION FOR POISSON, HELMHOLTZ AND ALLEN-CAHN PDEs

For every architecture, we disregard the parameters used in the first and last layers, as the input dimension of 2 and output dimension of 1 make them negligible compared to the number of parameters used in the intermediate layers. For all architectures and problems, we consider $\{1, 2, 4, 6\}$ intermediate layers (not considering first and last layers), as well as three different architecture-specific hyperparameters, as detailed below. **All model widths were then inferred to satisfy the model size at hand**. This results in each architecture having 12 possible hyperparameter configurations for each parameter size (4 for depth, times 3 for architecture-specific hyperparameter).

**Poisson & Helmholtz**: For ActNet, we use $\{8, 16, 32\}$ values of $N$ and for KAN grid resolutions of $\{3, 10, 30\}$. For Siren we consider $\omega_0$ in $\{\frac{\pi w}{3}, \pi w, 3\pi w\}$. For MLP we consider activations in $\{\texttt{tanh}, \texttt{sigmoid}, \texttt{GELU}\}$, where GELU is the Gaussian Error Linear Unit from (Hendrycks & Gimpel, 2020).

**Allen-Cahn**: For ActNet, we use $\{4, 8, 16\}$ values of $N$ and for KAN grid resolutions of $\{4, 8, 16\}$. For Siren we consider $\omega_0$ in $\{10, 30, 90\}$. For MLP we consider activations in $\{\texttt{tanh}, \texttt{sigmoid}, \texttt{GELU}\}$, where GELU is the Gaussian Error Linear Unit from (Hendrycks & Gimpel, 2020).

### G.3   POISSON

We consider the 2D Poisson PDE with zero Dirichlet boundary condition, defined by

$$\Delta u(x, y) = f(x, y), \qquad (x, y) \in [-1, 1]^2,$$
$$u(x, y) = 0, \qquad (x, y) \in \delta[-1, 1]^2.$$

We use forcing terms of the form $f(x, y) = 2\pi^2 w^2 \sin(\pi w x) \sin(\pi w y)$ for values of $w$ in $\{1, 2, 4, 8, 16\}$. Each of these PDEs has exact solution $u(x, y) = \sin(\pi w x) \sin(\pi w y)$, which are shown in Figure 6.

Each model was trained using Adam (Kingma & Ba, 2017) for 30,000 iterations, then fine tuned using LBFGS Liu & Nocedal (1989) for 100 iterations. We use a batch size of 5,000 points uniformly sampled at random on $[0, 1]^2$ at each step. For training using Adam, we use learning rate warmup from $10^{-7}$ to $5 \times 10^{-3}$ over 1,000 iterations, then exponential decay with rate 0.75 every 1,000 steps, and adaptive gradient clipping with parameter 0.01 as described in Brock et al. (2021).

Furthermore, to avoid unfairness from different scaling between the residual loss and the boundary loss, we enforce the boundary conditions exactly for all problems by multiplying the output of the neural network by the factor $(1 - x^2)(1 - y^2)$, in a strategy similar to what is outlined in Sukumar & Srivastava (2022).

The final relative L2 errors can be seen in figure 8, while the final residual losses can be seen in figure 9. An example solution is plotted in figure 7 and sample computational times are reported in table 7.

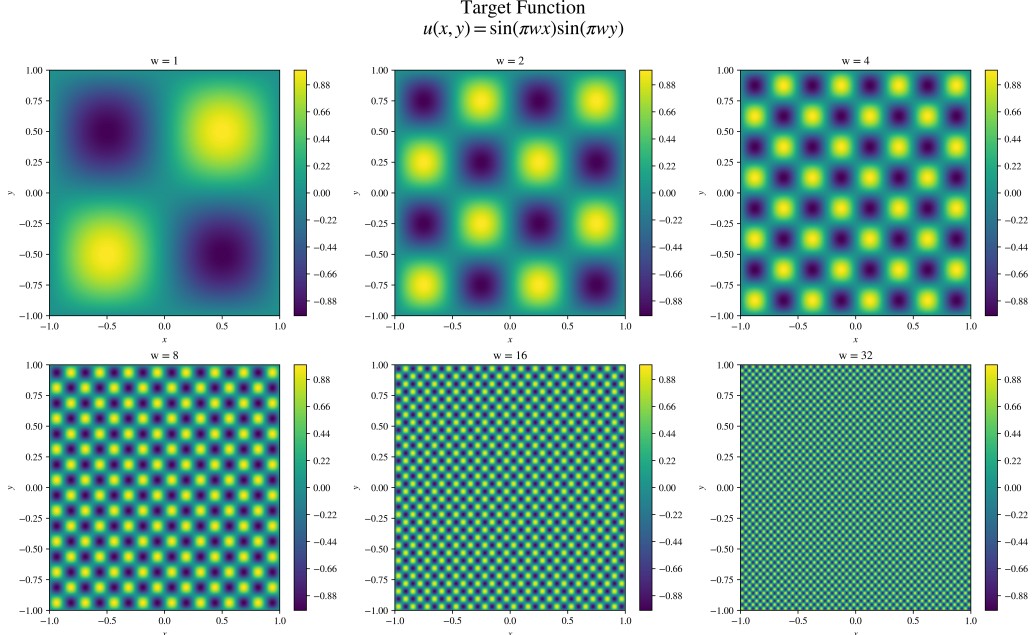

Figure 6: Visualization of the different target functions for the Poisson and Helmholtz PDEs. As the frequency $w$ increases, the magnitude of gradients become larger and the laplacian $\Delta u(x, y) = 2\pi^2 w^2 \sin(wx) \sin(wy) = 2\pi^2 w^2 u(x, y)$ scales as $\propto w^2$, thus creating a challenging problem for PINNs.

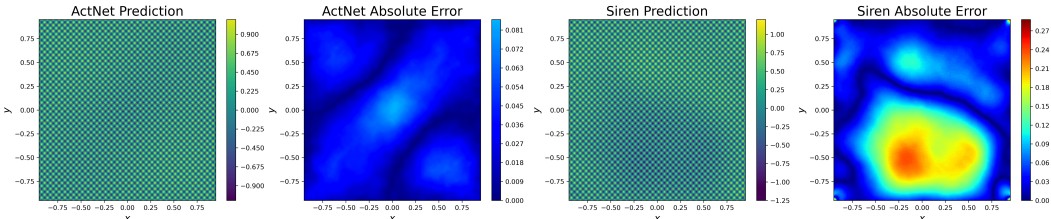

Figure 7: Example predictions for the Poisson equation using $w = 32$. The relative L2 errors for the ActNet, and Siren solutions above are 6.42e-02, 1.91e-1, respectively. Predictions for KANs are not plotted here, as they did not converge for this example.

### G.4 HELMHOLTZ

The inhomogeneous Helmholtz PDE with zero Dirichlet boundary condition is defined by

$$\Delta u(x, y) + \kappa^2 u(x, y) = f(x, y, ) \qquad (x, y) \in [-1, 1]^2,$$
$$u(x, y) = 0, \qquad (x, y) \in \delta[-1, 1]^2.$$

If we set $\kappa = 1$ and the forcing term to be $f(x, y) = (\kappa - 2\pi^2 w^2) \sin(\pi\omega x) \sin(\pi\omega y)$, then this PDE has exact solution $u(x, y) = \sin(wx) \sin(wy)$, which is pictured in figure 6.

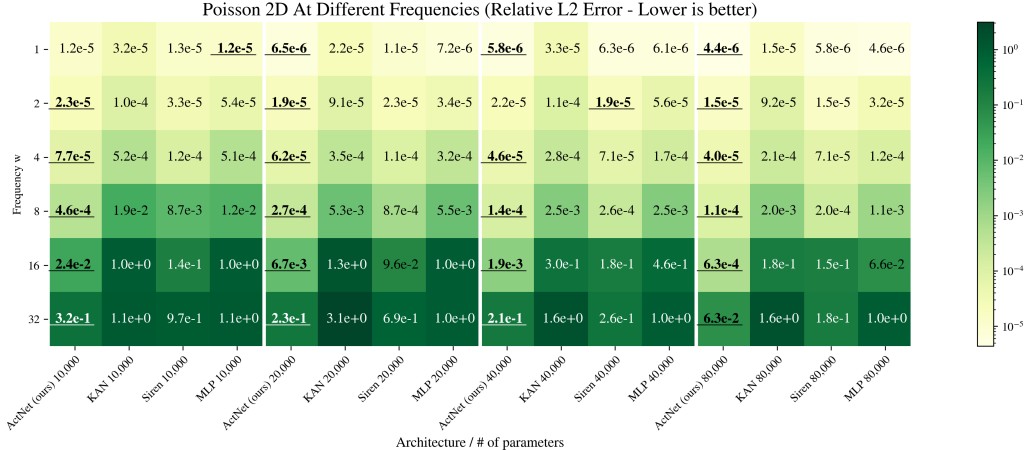

Figure 8: Results for the 2D Poisson problem using PINNs. For each hyperparameter configuration, 3 different seeds were used for initialization, and the median result was used. For each square, the best hyperparameter configuration (according to the median) is reported. The best performing method for each frequency $w$ and each number of parameters is underlined.

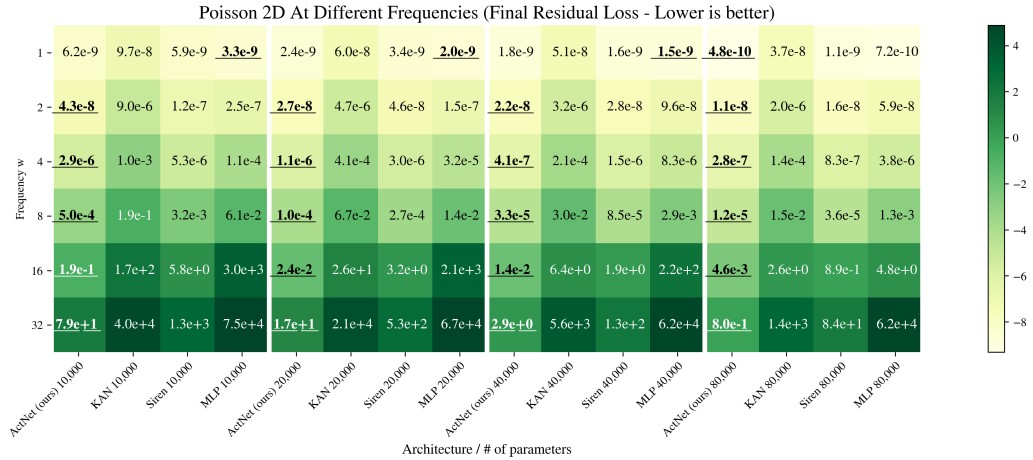

Figure 9: Final residual loss for the Poisson problem using PINNs. For each hyperparameter configuration, 3 different seeds were used for initialization, and the median result was used. For each square, the best hyperparameter configuration (according to the median) is reported. The best performing method for each number of parameters is underlined.

Each model was trained using Adam (Kingma & Ba, 2017) for 30,000 iterations, then fine tuned using LBFGS Liu & Nocedal (1989) for 100 iterations. We use a batch size of 5,000 points uniformly sampled at random on $[0, 1]^2$ at each step. For training using Adam, we use learning rate warmup from $10^{-7}$ to $5 \times 10^{-3}$ over 1,000 iterations, then exponential decay with rate 0.75 every

Table 7: Average computational time per Adam training iteration for the Poisson problem on a Nvidia RTX A6000 GPU. All times are reported in milliseconds. Each cell averages across three seeds and four different widths and depths hyperparameters as reported on G.2.

| Model Size | ActNet ($N = 8$) | ActNet ($N = 16$) | ActNet ($N = 32$) | KAN ($G = 3$) | KAN ($G = 10$) | KAN ($G = 30$) | Siren |
|---|---|---|---|---|---|---|---|
| 10k | 5.10 | 7.17 | 9.59 | 6.87 | 10.7 | 13.0 | 1.32 |
| 20k | 7.37 | 10.8 | 15.3 | 10.3 | 16.3 | 18.5 | 1.66 |
| 40k | 10.7 | 16.7 | 26.2 | 17.1 | 25.1 | 25.6 | 2.49 |
| 80k | 16.2 | 26.5 | 43.4 | 23.6 | 40.0 | 37.4 | 3.60 |

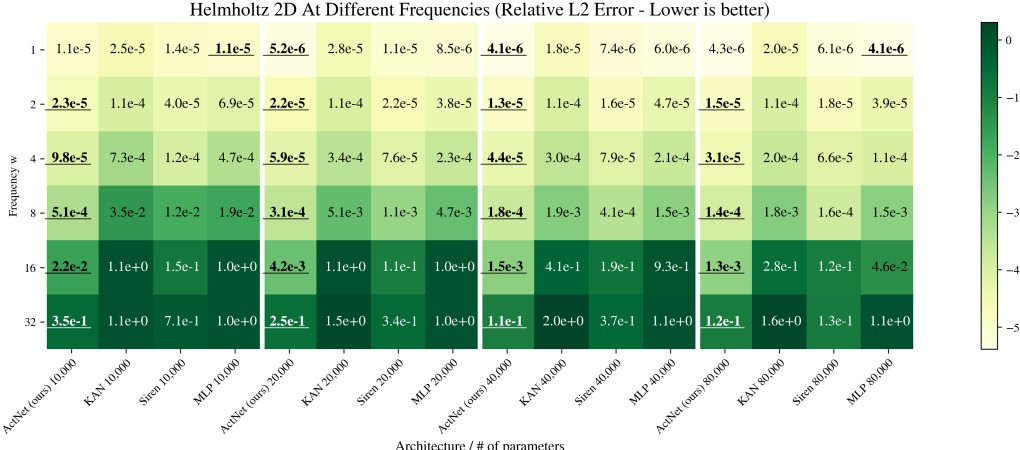

Figure 10: Results for the 2D Helmholtz problem using PINNs. For each hyperparameter configuration, 3 different seeds were used for initialization, and the median result was used. For each square, the best hyperparameter configuration (according to the median) is reported. The best performing method for each frequency $w$ and each number of parameters is underlined.

Table 8: Average computational time per Adam training iteration for the Helmholtz problem on a Nvidia RTX A6000 GPU. All times are reported in milliseconds. Each cell averages across three seeds and four different widths and depths hyperparameters as reported on G.2.

| Model Size | ActNet ($N = 8$) | ActNet ($N = 16$) | ActNet ($N = 32$) | KAN ($G = 3$) | KAN ($G = 10$) | KAN ($G = 30$) | Siren |
|---|---|---|---|---|---|---|---|
| 10k | 5.03 | 7.12 | 9.53 | 6.84 | 10.7 | 13.0 | 1.31 |
| 20k | 7.33 | 10.8 | 15.2 | 10.2 | 16.2 | 18.5 | 1.67 |
| 40k | 10.6 | 16.6 | 26.0 | 17.1 | 25.0 | 25.7 | 2.50 |
| 80k | 16.2 | 26.3 | 43.1 | 23.6 | 40.0 | 37.4 | 3.57 |

1,000 steps, and adaptive gradient clipping with parameter 0.01 as described in Brock et al. (2021). Furthermore, to avoid unfairness from different scaling between the residual loss and the boundary loss, we enforce the boundary conditions exactly for all problems by multiplying the output of the neural network by the factor $(1 - x^2)(1 - y^2)$, in a strategy similar to what is outlined in Sukumar & Srivastava (2022).

The final relative L2 errors can be seen in figure 10, while the final residual losses can be seen in figure 11. An example solution is plotted in figure 2 and sample computational times are reported in table 8.

### G.5 ALLEN-CAHN

We consider the Allen-Cahn equation with Dirichlet boundary and initial condition $u(x,0) = x^2 \cos(\pi x)$, defined as

$$
\frac{\partial u}{\partial t}(x,t) - D\frac{\partial^2 u}{\partial x^2}(x,t) + 5(u^3(x,t) - u(x,t)), \qquad (x,t) \in [-1,1] \times [0,1],
$$
$$
u(x,0) = x^2 \cos(\pi x), \qquad x \in [-1,1],
$$
$$
u(-1,t) = u(1,t) = -1, \qquad t \in [0,1].
$$

Each model was trained using Adam (Kingma & Ba, 2017) for 100,000 iterations and batch size of 10,000 points uniformly sampled at random each step on $[-1,1] \times [0,1]$. We use learning rate warmup from $10^{-7}$ to $5 \times 10^{-3}$ over 1,000 iterations, then exponential decay with rate 0.9 every 1,000 steps, stopping the decay once the learning rate hits $5 \times 10^{-6}$. We also use adaptive gradient clipping with parameter 0.01 as described in Brock et al. (2021). Furthermore, to avoid unfairness

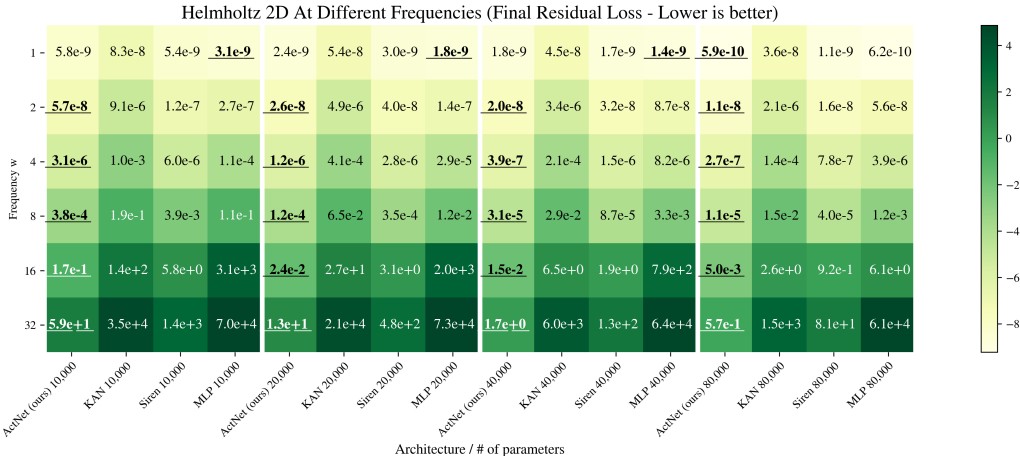

Figure 11: Final residual loss for the Helmholtz problem using PINNs. For each hyperparameter configuration, 3 different seeds were used for initialization, and the median result was used. For each square, the best hyperparameter configuration (according to the median) is reported. The best performing method for each number of parameters is underlined.

Table 9: Average computational time per Adam training iteration for the Allen-Cahn problem on a Nvidia RTX A6000 GPU. All times are reported in milliseconds. Each cell averages across three seeds and four different widths and depths hyperparameters as reported on G.2.

| Model Size | ActNet ($N = 4$) | ActNet ($N = 8$) | ActNet ($N = 16$) | KAN ($G = 4$) | KAN ($G = 8$) | KAN ($G = 16$) | Siren |
|---|---|---|---|---|---|---|---|
| 25k | 4.16 | 6.29 | 10.8 | 9.01 | 24.5 | 56.5 | 2.04 |
| 50k | 5.94 | 10.2 | 18.2 | 12.3 | 35.5 | 84.1 | 2.76 |
| 100k | 9.65 | 16.9 | 30.1 | 17.1 | 48.8 | 122.9 | 4.30 |

from different scaling between the residual loss and the boundary loss, we enforce the initial and boundary conditions exactly by setting the output of the model to $(1 - t)(x^2 \cos(\pi x)) + t[(1 - x^2)u_\theta(x, t) - 1]$, where $u_\theta$ is the output of the neural network, in a strategy similar to what is outlined in Sukumar & Srivastava (2022).

Finally, we also employ the causal learning strategy from Wang et al. (2024b), updating the causal parameter every 10,000 steps according to the schedule $[10^{-1}, 10^0, 10^1, 10^2, 10^3, 10^3, 10^3, 10^4, 10^4, 10^4]$.

The final relative L2 errors can be seen in figure 13, while the final residual losses can be seen in figure 14. An example solution is plotted in figure 12 and sample computational times are reported in table 9.

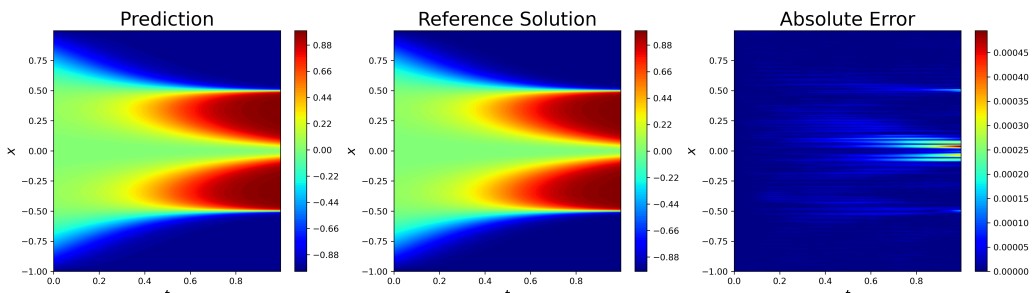

Figure 12: ActNet predictions for the Allen-Cahn equation. The relative L2 error is 4.51e-05.

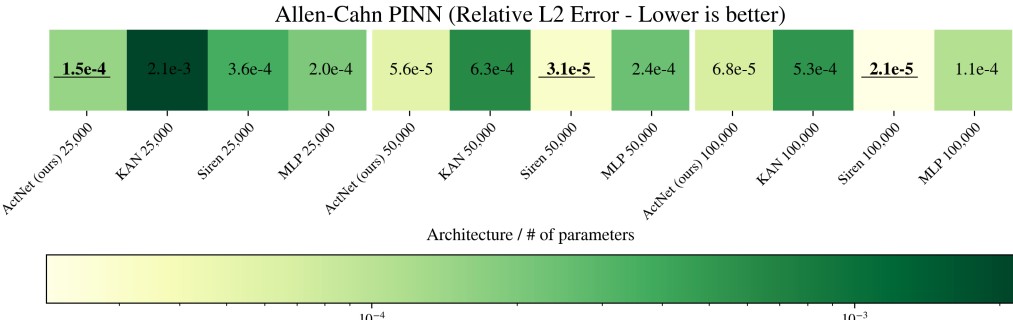

Figure 13: Results for the Allen-Cahn problem using PINNs. For each hyperparameter configuration, 3 different seeds were used for initialization, and the median result was used. For each square, the best hyperparameter configuration (according to the median) is reported. The best performing method for each number of parameters is underlined.

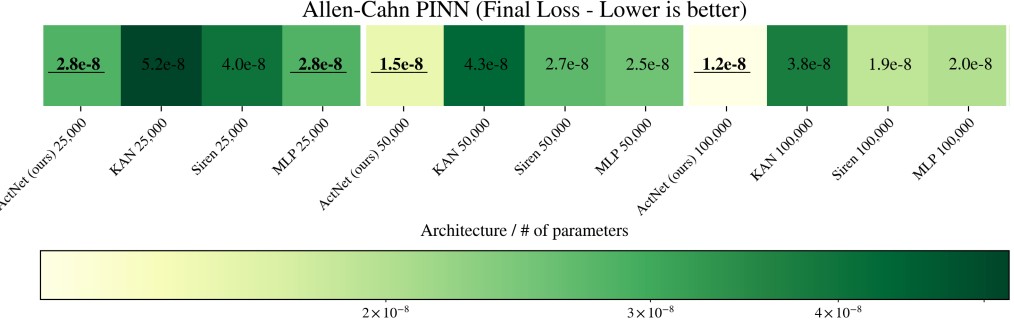

Figure 14: Final residual loss for the Allen-Cahn problem using PINNs. For each hyperparameter configuration, 3 different seeds were used for initialization, and the median result was used. For each square, the best hyperparameter configuration (according to the median) is reported. The best performing method for each number of parameters is underlined.

## G.6 ADVECTION

The 1D advection equation is defined by

$$\frac{\partial u}{\partial t} + c\frac{\partial u}{\partial x} = 0, \qquad t \in [0, 1], \ x \in (0, 2\pi),$$
$$u(0, x) = g(x), \qquad x \in (0, 2\pi),$$

with initial condition $g(x) = \sin(x)$ and transport velocity constant $c = 80$.

For our implementation, we use the JaxPi library from Wang et al. (2023; 2024a). We copy over the configuration of their `sota.py` config file, along with their learnable periodic embedding, changing the architecture to an ActNet with 5 layers, embedding dimension of 256 and $\omega_0$ value of 5. We train using Adam for 300,00 iterations with adaptive gradient clipping (Brock et al., 2021) with parameter 0.01. We use a starting learning rate of $10^{-3}$ and exponential rate decay of 0.9 every 5,000 steps. Each step took on average 40.0ms on a Nvidia RTX A6000 GPU.

For the PirateNet implementation, we used the code from the public GitHub repository `https://github.com/PredictiveIntelligenceLab/jaxpi/tree/pirate` and once again adapted the `sota.py` config file from (Wang et al., 2023), as was done with ActNet. For a comparable network size to the ActNet and modified MLP considered in (Wang et al., 2023), we set the network with two blocks of 3 layers each, totaling 6 layers, all with width 200. Each step took on average 11.9ms on a Nvidia RTX A6000 GPU.

## G.7 KURAMOTO–SIVASHINSKY

The Kuramoto–Sivashinsky equation we considered is defined by

$$\frac{\partial u}{\partial t} + \alpha u\frac{\partial u}{\partial x} + \beta\frac{\partial^2 u}{\partial x^2} + \gamma\frac{\partial^4 u}{\partial x^4} = 0 \quad t \in [0, 1], x \in [0, 2\pi]$$
$$u(0, x) = u_0(x) \quad x \in [0, 2\pi]$$

with periodic boundary conditions and constants $\alpha = 100/16$, $\beta = 100/16^2$ and $\gamma = 100/16^4$, along with initial condition $u_0(x) = \cos(x)(1 + \sin(x))$.

For our implementation, we use the JaxPi library from Wang et al. (2023; 2024a). We copy over the configuration of their `sota.py` config file, changing the architecture to an ActNet with 5 layers, embedding dimension of 256 and $\omega_0$ value of 5. We train using Adam (this time without adaptive gradient clipping) using a starting learning rate of $10^{-3}$ and exponential rate decay of 0.8 every 3,500 steps.

We split the time domain $[0, 1]$ into 10 windows of equal length. Since the precision of the solution matters most in the initial time steps (most of the error from the latter windows is due to propagated error in the initial conditions), we train the first window for 250k steps, the second and third for 200k steps, and all other for 150k steps. This results in 15% less steps than what was done in JaxPi, where each window is trained for 200k steps. Each step took on average 66.7ms on a Nvidia RTX A6000 GPU.

For the PirateNet implementation, we used the code from the public GitHub repository `https://github.com/PredictiveIntelligenceLab/jaxpi/tree/pirate` and once again adapted the `sota.py` config file from (Wang et al., 2023), as was done with ActNet. For a comparable network size to the ActNet and modified MLP considered in (Wang et al., 2023), we set the network with two blocks of 3 layers each, totaling 6 layers, all with width 200. Each step took on average 22.2ms on a Nvidia RTX A6000 GPU.

