# OpenReview forum: "Deep Learning Alternatives Of The Kolmogorov Superposition Theorem"
_ICLR.cc/2025/Conference — ICLR 2025 Spotlight_

### Official Review · Reviewer_CpiV · 2024-11-01

**Soundness:** 4
**Presentation:** 3
**Contribution:** 3
**Rating:** 8
**Confidence:** 4

**Summary:**

The paper proposes ActNet, a novel neural network architecture based on Kolmogorov Superposition Theorem (KST), as an alternative to Kolmogorov-Arnold Networks (KANs). The authors use the Laczkovich version of KST to develop their architecture instead of the original KST emplyoed in KANs. They prove universal approximation properties and propose an initialization scheme for their method which they assures stability of network activations.

Empirically, they evaluate their model on PDE simulation tasks using a PINN framework. They compare their model to KANs and MLPs on a range of PDE equations such as Poisson, Helmholtz, Allen-Cahn and Kuramoto-Sivashinsky. Their method has comparable accuracy to the benchmarks chosen for comparison.

**Strengths:**

1. I find the paper to be well-written. The exposition of KST and of the ActNet architecture are clearly presented. The contributions made by this work is also clearly outlined.
2. As far as I am aware, the technical contributions in this paper are novel. The proposed architecture outperforms other similar models (KST variants such as KANs and MLPs) on the chosen tasks, has better parameter efficiency and seems to benefit from good theoretical guarantees (universal approximation properties for a two layer network).
3. I found the experimental section of the paper to be strong. Information about implementation and the considered PDEs is clearly given either in the main paper or in the appendix. Multiple ablations are conducted to validate the model. The choice of experimental framework is well motivated and the code is publicly available on Github.

All in all, I find this paper presents an interesting alternative to KANs.

**Weaknesses:**

1. Although interesting when compared to KANs and other MLPs, this method is not competitive when compared to SoTA ML informed solvers which use more sophisticated inductive biases such as [1,2,3,4] for example.
2. I find the theoretical contribution of the paper to be a bit weak. Although an interesting result, it seems trivial to me that a method based on Superposition Theorems for representing multivariate functions would benefit from universal approximation properties. Moreover, it is not clear to me why the property presented in Theorem 3.4 is a selling point of your method. Even if this property didn't hold, we could simply use some form of LayerNorm to enforce this. Also, in the context of modelling physical equations, it is not clear to me that inputs would be distributed following a $(0,1)$ Normal distribution in the first place.
3. There are a few typos/minor mistakes in the proofs. I will list them by point below.

Comments about the proof of Theorem 3.3:
* Line 868: typo, I believe you meant $poly_g$ not $pol$.
* "by making $\sum \lambda \phi (x)$ sure that is at most $δ$ far from the approximation $\sum \lambda \hat{\phi} (x)$". You do not introduce/define this approximation before mentioning it here, I think it would make this step clearer if you defined what this approximation is.

Comments about the proof of Theorem 3.4
* "After properly initializing the $β$ and $λ$ parameters (detailed in Appendix F)" this is not detailed in the appendix. Thus, when defining the expectation and second moment of these variables, it is not clear where the values you obtain come from (around lines 981 and 1006 )
* Do you consider the that $p=0$ and $\omega \sim \mathcal{N}(0,1)$ for this proof? If so, the eq. at line 984 is not clear to me. Same for the step between lines 1006 and 1009.

Also here are some minor comments you may want to address for the final version:
* "Table 1" not "table 1" (line 67 for instance).
* $m$ in Table 1 is not described until section 3. It would be nice to know (at least in big O) what this value is in terms of $d$ when looking at this table.
* $\beta$ should be bolded since it is a matrix, not a scalar (line 237).
* Inconsistent use of $\epsilon$ and $\varepsilon$ (around line 295 theorem 3.3 def 3.2).
* The grid size considered for the experiments is not reported in main paper and generally hard to find.
* The "Discussion" section appears to be more of a conclusion to me than a discussion of results.

[1] Li, Z., Kovachki, N., Azizzadenesheli, K., Liu, B., Bhattacharya, K., Stuart, A., & Anandkumar, A. (2020). Fourier neural operator for parametric partial differential equations. arXiv preprint arXiv:2010.08895.

[2] Gin, C., Lusch, B., Brunton, S. L., & Kutz, J. N. (2021). Deep learning models for global coordinate transformations that linearise PDEs. European Journal of Applied Mathematics, 32(3), 515-539.

[3] Lusch, B., Kutz, J. N., & Brunton, S. L. (2018). Deep learning for universal linear embeddings of nonlinear dynamics. Nature communications, 9(1), 4950.

[4] Li, Z., Kovachki, N., Azizzadenesheli, K., Liu, B., Stuart, A., Bhattacharya, K., & Anandkumar, A. (2020). Multipole graph neural operator for parametric partial differential equations. Advances in Neural Information Processing Systems, 33, 6755-6766.

**Questions:**

* The original KAN paper boasts interpretability of the learnt network. Does your method offer any interesting results in terms of interpretability?
* Do you have any experiments out-of-distribution (OOD) regime? It would be interesting to see how the model reacts to change in initial conditions or boundary conditions for example.
* Why didn't you test on the function approximation tasks as presented in the original KAN paper? Given your method is a direct competitor to the KAN architecture, such results would be interesting to have.
* How does your method perform given a different function basis?
* Are $p$ and $\omega$ trainable parameters?
* My understanding of the proof of Theorem 3.3 is that you use Weierstrass theorem to create polynomial bases which approximate both i) the sum of $g$s ii) the weighted sum of $\phi$s up to the desired precision $\epsilon$. However, the link between this and the architecture is not immediately clear to me.

    From what I gather, each polynomial approximation can be represented by an ActLayer with a polynomial basis of size $N_g$/$N_q$ and well chosen $\beta$ values. Is my reasoning correct here? I think it would improve clarity if you mentioned how your proof relates to the proposed architecture.

---

> ### Author Response · Authors · 2024-11-21
> **Author's Response (part 1)**
>
> First of all, thank you for your careful review and comments! We deeply appreciate them and hope to address your main points below:
> - **Comparison with State-of-the-Art Methods**: We appreciate the references to foundational work in neural operators and have made sure to cite them in the paper. While these methods demonstrate impressive capabilities in data-driven PDE solving, our work deliberately focuses on the physics-informed learning paradigm where solutions are learned without access to training data. This choice is motivated by several factors:
>   1) The unique challenges of physics-informed learning, where models must learn to satisfy differential operators through residual minimization rather than direct supervision.
>   2) The fundamental differences in problem formulation between operator learning (learning solution maps across PDE parameters) and our focus on high-accuracy solutions for specific PDEs.
>   3) The opportunity to establish ActNet's capabilities in a well-defined setting before extending to hybrid frameworks.
>
>   To ensure rigorous evaluation within the physics-informed framework, we have expanded our comparisons to include PirateNet, currently considered state-of-the-art for PINNs due to its sophisticated inductive biases and demonstrated performance across numerous benchmarks. Our results show that ActNet achieves superior performance on challenging test cases including the Advection equation and Kuramoto-Sivashinsky system. Notably, ActNet accomplishes this without employing PirateNet's advanced components like physics-informed initialization or adaptive residual connections.
>
>   Regarding extension to neural operators, we are actively investigating principled approaches to incorporate ActNet into operator learning frameworks. Rather than simply replacing MLPs with ActNet blocks, we believe the key challenge lies in designing appropriate conditioning and modulation mechanisms that preserve ActNet's theoretical foundations from Kolmogorov theory while enabling parameter-dependent solution maps. This represents an exciting direction for future work that could potentially combine the strengths of both physics-informed and data-driven paradigms.
>
> - **Theoretical Contributions**:
>   - **Relevance of Universal Approximation**: While we agree that it is not necessarily surprising that ActNet has universality properties, this serves as a good “sanity check” when designing new neural network architectures. It is true that a network with universal approximation might not perform well, but a network without this guarantee is most likely doomed to failure. As such, our universality statement gives a basic theoretical guarantee, in addition to presenting alternative perspectives into the way in which a neural network can approximate multivariate functions.
>   - **Relevance of Theorem 3.4**: This theorem tells us that the activations of an ActNet will not vanish or blow up to large values at initialization. It differs from LayerNorm in the sense that it guarantees that for a single neuron, activations will be roughly distributed as a standard normal *across different inputs to the network*, while LayerNorm enforces that for a single input, activations are normalized *across different neurons in the same layer*. In this sense, our approach is in fact closer to a BatchNorm layer. However, the stochastic nature of BatchNorm, where the normalization factor changes depending on unrelated inputs from the same batch, effectively prevents its integration with methods that hope to achieve very high accuracy for PDEs, as the added noise perturbs the inner layers of the network. Being able to compute this normalization in closed-form as we have done for ActNet, without depending on given inputs in a mini-batch, yields further stability, which allows for better training, as indicated in our experiments. We hope this helps clarify your question!
>   - **Distribution of Inputs**: The assumption that inputs to a neural network are normalized is common in many deep learning papers, and in fact most practitioners recommend normalizing datasets so that each entry has mean 0 and variance 1, which is also what we have done in our experiments. Additionally, since we are composing several ActLayers sequentially in an ActNet and we know that outputs of an ActLayer are roughly normally distributed, this construction becomes useful not only for the initial ActLayer, but also for every subsequent one.

---

> > ### Author Response · Authors · 2024-11-21
> > **Authors' Response (part 2)**
> >
> > - **Typos/Minor Mistakes in Proofs**:
> >   - **Theorem 3.3**: You are correct about both these typos. $\hat{\phi}_q$ should be $poly_q$ as well. We have now corrected them in the proof.
> >   - **Theorem 3.4**:
> >     - The initialization scheme is detailed in appendix D.3, not F as originally said. We have now corrected this typo.
> >     - In the proof for theorem 3.4, all the mentions of $\sin(\omega_j x_i + p_j)$ should have been $b_j(x_i)$ instead (these two differ by an affine transformation as computed in Lemma F.1, which makes $\mathbb{E}\left[b_j(x_i)\right]=0$ as claimed, no matter the choice of $p$ and $\omega$). Thank you for catching this typo! This mistake would make a big difference for the correctness of proof. We hope these corrections help clarify some of your questions about the proof, and we are available to discuss more if there are any other concerns.
> > - **Minor Comments**: (thank you for the sharp eye!)
> >   - Overall, all the typos/inconsistencies you pointed out were indeed mistakes on our part and we have incorporated the corrections into the paper. We appreciate you for catching them!
> >   - We train all networks using random points sampled uniformly on the input domain, taking advantage of the mesh-less nature of PINNs. Because of this, we don’t have a fixed grid size, but rather a fixed batch-size for each iteration.
> > - **Interpretability**: KAN employs interpretability by symbolically regressing their inner functions to find the closed-form function that best approximates it. While doing this is possible with ActNet as well, we do not implement this functionality ourselves, due to implementation time constraints. Additionally, for most realistic applications in scientific computing, there are no closed-form solutions to a PDEs, thus limiting the possibility for interpretation past very simple problems.
> > - **Out-of-distribution Regime**: As we have not trained ActNet on data-driven problems, it is unclear what its behavior would be on out-of-distribution points, since when training using PINNs we train on the entire domain, leaving no point outside of the distribution. We agree, however, that in the case of solving PDEs using operator learning, for example, conducting OOD experiments would be a critical and very interesting experiment to conduct. We intend to explore this direction in future work, once we extend ActNet experiments to data-driven and operator learning problems.
> > - **Function Approximation Tasks from KAN paper**: Many of the approximation tasks presented in the KAN paper are for very smooth functions, where most methods are expected to perform very well, approaching the limit of what is possible using neural networks with single-precision floating point operations. The original KAN paper in fact also considers the Poisson 2D problem, but with a slightly different forcing function, which is similar to our benchmark using $w=1$ or $w=2$. We believe these smooth benchmarks are likely not indicative of performance for realistic tasks, and instead decided to focus on more challenging problems. We also decided to focus this paper on PINNs, which require no data, as opposed to data-driven tasks, leaving this subject for future work.
> > - **ActNet basis functions**: During initial exploratory phases of our work, we also explored using polynomial basis functions in the implementation of ActNet. Overall, this approach also worked, but slightly less robustly than using the sinusoidal basis. For the sake of simplicity, we decided to stick to a single type of basis function, but as future work we may include other basis possibilities as well.
> > - **$p$ and $\omega$ parameters**: It is possible, but not necessary in most cases, to train the frequencies and phases. While we chose to let these parameters be trainable in all experiments reported in the paper, in our internal experiments we do not find much difference in performance by fixing them at initialization. We have a short discussion on training these parameters in Appendix D2.
> > - **Relationship Between Proof and Architecture**: You are correct about your understanding of the proof: we approximate both the inner functions and the outer function to good enough accuracy that the overall representation is also close enough to the true expression given by the theorem. Regarding the relationship between the theorem and our architecture, this proof serves as a basic “sanity check” for our method, as described above in the bullet point for Weakness 2. In fact, in our newly included appendix D.4 detailing hyperparameter choices, we see that we need at least two ActLayers to obtain good performance, which is indicated by the theorem, since we need at least two layers to obtain universal approximation guarantees.

---

> > > ### Comment · Reviewer_CpiV · 2024-11-23
> > >
> > > Thank you for the detailed response, I have read it and the other reviews carefully.
> > >
> > > In light of your response, and, in particular due to
> > > * the justification of the proposed benchmarks,
> > > * the addition of PirateNet to the benchmarks
> > > * The clarifications/corrections concerning the proofs
> > >
> > > I am modifying my recommendation from "weak reject" to "accept."

---

### Official Review · Reviewer_3BDJ · 2024-11-03

**Soundness:** 4
**Presentation:** 4
**Contribution:** 4
**Rating:** 8
**Confidence:** 5

**Summary:**

This paper investigates alternative approaches to the Kolmogorov Superposition Theorem (KST) for neural network design, addressing practical limitations of the original KST formulation, such as its complexity and lack of structural insights. Kolmogorov-Arnold Networks (KANs) utilize KST for function approximation but have shown inconsistent performance compared to traditional multilayer perceptrons (MLPs). The authors introduce ActNet, a scalable model that builds on KST while mitigating some of its original limitations. Evaluated within the Physics-Informed Neural Networks (PINNs) framework for PDE simulation, ActNet consistently outperforms KANs and competes well with leading MLP-based methods, marking it as a promising direction for KST-based deep learning in scientific computing.

**Strengths:**

1. **Originality**: The paper brings a fresh perspective to neural network design by leveraging alternative formulations of the Kolmogorov Superposition Theorem (KST). Introducing ActNet, based on Laczkovich's theorem, reflects a novel approach to overcoming the limitations of Kolmogorov-Arnold Networks (KANs), making KST more applicable to practical deep learning tasks, particularly within Physics-Informed Neural Networks (PINNs).

2. **Quality**: The research is thorough, with ActNet being tested across multiple benchmarks against established models like KANs and MLPs, demonstrating its advantages in function approximation and handling PDE simulations. Theoretical foundations are strong, with proofs of ActNet’s universal approximation properties, and empirical results consistently show ActNet’s improved performance in accuracy and stability.

3. **Clarity**: The paper is well-structured, presenting a clear motivation for the need for alternative KST formulations, followed by detailed descriptions of ActNet’s architecture and its theoretical underpinnings. The explanations of mathematical concepts and the positioning of ActNet within current scientific computing challenges are accessible and well-supported with illustrative figures and tables.

4. **Significance**: ActNet addresses critical limitations in applying KST to neural networks, opening new possibilities for KST-based models in scientific computing and PDE simulation. The model's competitive performance against leading MLP-based approaches highlights its potential impact on advancing scientific machine learning, particularly in low-dimensional function approximation and complex simulations, where existing architectures struggle.

**Weaknesses:**

1.  Although ActNet performs well on PINNs, its comparisons are limited to specific benchmarks and do not consistently compare against the latest models for PINNs.
2. The paper lacks detailed ablation studies on critical design choices within ActNet, such as the basis functions used or the impact of ActLayer depth. Including these analyses would clarify the sensitivity of ActNet’s performance to these parameters and help guide future implementations or adaptations of the model.
3. While the paper offers rigorous theoretical grounding, some sections—particularly those detailing the inner workings of the ActLayer—may be dense for readers unfamiliar with KST.

**Questions:**

1. As you mentioned about JAXPI, Why you didn't compare with their latest result which is Piratenet [1] that has been pubilshed Feb 2024 rather than Causal PINN that from 2022. Also I saw that you have cited this paper as well.
2. What happen if you apply ActNet into more chaotic system, for example Navier–Stokes equations. is ActNet still performs better?
3.

[1] PirateNets: Physics-informed Deep Learning with Residual Adaptive Networks

---

> ### Author Response · Authors · 2024-11-21
> **Authors' Response**
>
> First of all, we thank you for your careful review and comments! We deeply appreciate them and hope to address your main points below:
> - **Comparison against PirateNet**: That is an excellent point, thank you for bringing it up! The reason we didn’t originally include comparisons against PirateNet in the original manuscript is because the paper did not run these specific benchmarks. Having said that, you are absolutely right that PirateNet is another important competitive method to compare against, and we have since ran the advection and KS equations using a PirateNet and have confirmed that ActNet still generally outperforms PirateNet in these benchmarks. More specifically, ActNet significantly outperformed PirateNet both in the advection equation and the full time solution of the KS equation, but very slightly underperformed PirateNet (1.3e-5 vs 1.2e-5) when considering only the first time window, which is fairly smooth and easier to get good results on. ActNet particularly shines in the KS equation for later times, where the chaotic nature of the system is more apparent.
>
>   Here we should also note that, additionally, PirateNet utilizes a physics-informed initialization and an adaptive residual connective structure that is also compatible with ActNet. Although we leave this integration as future work, we believe that using this strategy would further improve the performance of ActNet, making it even more competitive against PirateNet.
>
> - **Ablation on ActNet hyperparameters**: You are correct that we have overlooked including detailed information about ablations with hyperparameter tuning in the manuscript. This is something we have done but did not report in the original manuscript. Here we take the opportunity to revise our Appendix and include the test error for all hyperparameter combinations considered for ablation experiments. In short, we find that the basis size $N$ does not need to be large for accurate results. In fact, except for small parameter counts, we don’t see accuracy gains for increasing the value of $N$ beyond 4 or so, while the computational complexity of the model increases for larger $N$ (even when network size is fixed). Additionally, as is often the case for deep learning, we find that composing more layers and increasing network width generally improves performance by increasing network capacity. We also have all over 7,000 runs logged on WeightsAndBiases, and plan on making them publicly available, including loss curves, if the paper is accepted (we currently cannot do this, as there is no way of anonymously sharing data using the WeightsAndBiases platform).
> - **Dense Theoretical Sections**: While we did our best to present the mathematical details in a clear and understandable manner, we understand some derivations might be dense for some readers. However, we believe these mathematical insights are a key contribution of our paper, and the 10 page maximum limits our ability to further extend some explanations. Having said that, we plan to produce other material later to accompany this paper, including detailed blog posts and video explanations. While we regret there is not enough space for more detailed exposition in the paper, we hope this helps readers parse the material in the future.
> - **Comparison with State-of-the-Art Methods**: As discussed above, we have conducted comprehensive comparisons against PirateNet, demonstrating ActNet's strong performance on challenging benchmarks. Our original comparisons utilized the JaxPi framework with all optimizations from Wang et al. (2023) "An Expert's Guide to Training Physics-Informed Neural Networks." ActNet's implementation incorporates several advanced techniques from this work, including causal learning strategies for all time-dependent PDEs (Allen-Cahn, advection, and Kuramoto–Sivashinsky equations). These methodological choices ensure fair comparison against current state-of-the-art approaches while highlighting ActNet's inherent architectural advantages. The fact that ActNet achieves superior performance even without some of PirateNet's sophisticated components (such as physics-informed initialization) further validates our theoretical framework.
> - **ActNet on Chaotic Systems**: That is a great question! The Kuramoto-Sivashinsky (KS) experiment considered in section 4 is considered a chaotic problem for the choice of PDE parameters we used, and we observe performance above what can currently be found in the literature. We have not thoroughly ran comparisons using the Navier-Stokes equation, but believe ActNet can perform well under this setting as well. We will do our best to include an additional benchmark using the Navier-Stokes equation before the rebuttal period is over, but we are currently limited by implementation/computational time constraints.

---

> > ### Author Response · Authors · 2024-11-30
> >
> > Hello! We wanted to post a reminder of the end of the extended discussion period, which is ending very soon. We have posted a rebuttal and did our best to answer the reviewers' questions and concerns, almost all of which we believe were successfully addressed (see the reply to your review + the "Summary of Improvements" post). We make ourselves available for further discussion, and to answer any remaining questions you might have, but it would be helpful to have enough time to properly respond. We would appreciate an updated response, as we reach this critical point of the reviewing process.

---

> > > ### Comment · Reviewer_3BDJ · 2024-12-02
> > >
> > > Dear authors,
> > > Thank you for addressing the comments and suggestions provided during the review process. I appreciate the effort and thoughtfulness you have put into revising the manuscript. I will raise the marks.

---

### Official Review · Reviewer_JpDB · 2024-11-04

**Soundness:** 3
**Presentation:** 3
**Contribution:** 2
**Rating:** 6
**Confidence:** 4

**Summary:**

Based on a new formulation of the Kolmogorov Superposition Theorem (KST), this paper introduces a neural network ActNet to solve PDEs. ActNet is proposed as a global approximation in solving PDEs, which is similar to PINN. The authors demonstrate ActNet's theoretical properties, including universality, and validate it on a range of PDE benchmarks, showing improved performance over KAN and SIREN.

**Strengths:**

1. The authors provide a solid theoretical foundation for ActNet, detailing its universal approximation capabilities and presenting a well-motivated formulation based on KST. This paper compares the complexity of different formulations well.
2. Three 1D examples and two 2D examples are included to demonstrate that ActNet can achieve better performance.
3. This paper compares the model performance with other open-source JaxPi frameworks to enhance reproducibility and fairness.

**Weaknesses:**

1. The selected examples primarily use sinusoidal forcing terms, including equations like Poisson, Helmholtz, and Allen-Cahn. The proposed model uses sine functions as basis functions, which justifies its improved performance over the spline basis functions used in KAN. In Figure 12, we can see that SIREN can have the best performance. Therefore, comprehensive benchmarks should be included for challenging 2D and 3D problems, including the Navier–Stokes equations and turbulence cases.
2. As this paper emphasizes the comparison between MLP and KAN, it is essential to include a general MLP model for performance evaluation alongside SIREN.

**Questions:**

1. This paper presents a novel approach for solving PDEs, similar to PINN. However, since the PDEs are known, why not use traditional numerical methods, such as FEM or FVM, on GPUs? A fair comparison is needed, focusing on accuracy, efficiency, and implementation complexity.
2. How does the proposed method address high-dimensional input problems?
3. Line 161: Why is the pathological behavior of the inner function relevant to the exact representation rather than the approximation?
4. Table 2: A relative L2 error should suffice for accuracy comparison, so why include the residual loss? Additionally, why does the Allen-Cahn example in Table 2 lack a consistently best-performing model?

---

> ### Author Response · Authors · 2024-11-21
> **Authors' Response (part 1)**
>
> First of all, we thank you for your careful review and comments! We deeply appreciate them and hope to address your main points below:
> - **Sinusoidal Basis**: While it is true that for the 2D Poisson and Helmholtz problems used as ablation the sinusoidal forcing terms make Fourrier-like architectures better suited, the Allen-Cahn problem with Dirichlet boundary, as well the Kuramoto-Sivashinsky PDEs do not, displaying very sharp transitions in the ground truth solution for later times. These sharp transitions often present a challenge for Fourier-like methods, due to Gibbs phenomena, but are remarkably not an issue for the ActNet architecture, as can be seen in the solutions plotted in Figures 3, 4 and 12. Additionally, we also compare against Siren specifically to consider an MLP baseline that is well-suited for this type of problem. Overall, we find that ActNet generally outperforms other methods, both for sinusoidal-like problems and not.
> - **Comparison against MLP baselines**: Initially, we only used Siren as a baseline for the MLP for fairness, since, as you mentioned, some of the forcing terms/initial conditions we used are sinusoidal, so we felt that Siren would present the most realistic/best performing baseline for MLP. Having said that, you are absolutely right that it is also worthwhile to include a non-sinusoidal MLP, so we have now ran tests using more conventional MLPs (with tanh, gelu and sigmoid activations) and included them in the paper. Overall, the performance trend resulted in roughly what we originally expected: ActNet > Siren > Traditional MLPs > KAN, although surprisingly we found that traditional MLPs sometimes overperformed Siren (see the updated table 2). Overall, we still see ActNet outperform alternative methods, especially in the more challenging problems with high frequency content. You can see these results in our experiments section, as well as in appendix G.
> - **Comparison against classical methods**: While established numerical methods (e.g., FEM/FVM) excel at solving forward PDE problems, the deep learning approach offered by PINNs presents unique advantages that warrant continued research and development. First, PINNs are particularly well-suited for inverse problems and parameter identification tasks, where traditional methods often struggle. Second, once trained, PINNs can enable rapid inference across different parameter configurations without requiring additional solving - a capability especially valuable for real-time applications and parameter studies. Third, PINNs' meshless nature allows them to handle irregular geometries and adaptive resolution naturally. Our work with ActNet advances the PINN framework by introducing a more theoretically-grounded architecture that demonstrates improved accuracy and stability. While we acknowledge classical methods' current superiority for standard forward problems, we believe innovations like ActNet are crucial steps toward realizing the full potential of physics-informed deep learning for a broader class of scientific computing challenges.
> - **High dimensional inputs**: The theoretical foundations of ActNet through the Kolmogorov Superposition Theorem (KST) extend naturally to arbitrary dimensions, potentially making our architecture well-suited for high-dimensional problems. In fact, our formulation based on Laczkovich's theorem scales linearly with input dimension ($\mathcal{O}(d)$), unlike the original KST's quadratic scaling ($\mathcal{O}(d^2)$) used in KANs. We strategically focused our initial investigation on low-dimensional domains (2-4D) for several reasons: (1) these dimensions encompass many important scientific computing applications including PDEs, neural fields, and physics simulations; (2) they allow rigorous empirical validation of our theoretical guarantees; and (3) they provide a clear benchmark against existing methods. The success of ActNet in this regime, particularly for challenging problems like the Kuramoto-Sivashinsky equation, suggests promising scalability to higher dimensions – a direction we are actively exploring for future work.
> - **Exact vs approximate representation**: In the case of exact representation, inner functions are continuous, but may present an infinite set of discontinuities. By requiring only approximation as opposed to exact representation, however, it might be possible to use smoother inner functions. Of course, in the limit as we require the approximation to be infinitely accurate, we might still run into inner functions that are differentiable, but have very high modulus of continuity. However, we also conjecture that the added compositional structure of ActNet, where several layers are computed in sequence may further alleviate some of these theoretical constraints, allowing for efficient representation using smooth inner functions. We believe our experiment section provides some evidence towards this conjecture.

---

> > ### Author Response · Authors · 2024-11-21
> > **Authors' Reponse (part 2)**
> >
> > - **Reporting Residual Loss**: While relative L2 error provides the primary measure of solution accuracy, we include PDE residual loss for several important reasons: (1) it directly quantifies how well the learned solution satisfies the underlying differential equation across the entire domain; (2) it offers insights into optimization dynamics during training; and (3) it helps diagnose potential issues with constraint satisfaction that may not be captured by L2 error alone. For the Allen-Cahn equation specifically, we observe that all methods achieve remarkably low errors ($\approx 10^{-5}$), approaching the theoretical limits of single-precision floating point arithmetic. At this scale, subtle numerical effects become significant, including:
> >   - Stochastic variations from mini-batch sampling
> >   - Initialization-dependent optimization paths
> >   - Round-off error accumulation in gradient computations
> >   - Floating point instabilities in higher-order derivatives
> >
> > To ensure robust evaluation, we employ a rigorous protocol of running each hyperparameter configuration with 3 distinct random seeds and reporting median performance. This methodology allows us to distinguish genuine architectural advantages from numerical artifacts, while acknowledging the fundamental precision limitations inherent in neural PDE solvers.

---

> > > ### Comment · Reviewer_JpDB · 2024-11-27
> > >
> > > Thank you for the authors' responses. Based on your clarifications, I have slightly adjusted the rating. However, my main concern remains: demonstrating promising results for high-resolution, challenging 3D cases and ensuring fair comparisons with classical numerical methods are critical. While PINN-related methods have their advantages, as the authors noted, the high error relative to classical numerical methods currently limits their applicability in real-world, high-stakes scenarios.
> > >
> > > This paper also suffers from the weak baseline issue, as discussed by McGreivy and Hakim (Weak baselines and reporting biases lead to overoptimism in machine learning for fluid-related partial differential equations. Nat Mach Intell, 6, 1256–1269, 2024).
> > >
> > > Despite these limitations, this paper does have contributions to KAN-related research directions and has good writing and rigid mathematics proofs.

---

> > > > ### Author Response · Authors · 2024-11-27
> > > >
> > > > First of all, thank you for the comments and updated review!
> > > >
> > > > Regarding some of the limitations you presented, while we agree that classical numerical methods for PDEs currently outperform machine learning based ones, and that papers that claim otherwise generally compare against very weak classical baselines (as discussed in McGreivy & Hakim), we still believe there is reason to continue exploring the limit of what ML approaches can achieve. As this field is, generally speaking, less than a decade old, improvements occur fast, particularly in the past 5 years, both in terms of predictive accuracy and the type of problems that can be approached using neural networks. In addition to this, some specific PDE-related tasks can already benefit from ML approaches, for example, in the case of inverse problems and problems that require low-latency predictions, as a neural network can be trained (slowly, compared to classical methods) ahead of time, then carry out new predictions in just a few milliseconds, even for challenging problems.
> > > >
> > > > We also believe that our benchmarks present challenging problems, and our predictive accuracy borders the limit of what can realistically be achieved with single-precision computing in many cases (relative L2 errors in the order of 1e-5 or 1e-6). Particularly, the Kuramoto-Sivashinsky PDE is a chaotic problem where a lot of ML-based approaches struggle significantly. We also run very extensive comparisons against other deep-learning based approaches, in order to thoroughly validate the efficiency of our method over KANs and MLP-based architectures.
> > > >
> > > > Having said this, your comments are very valid, and the current limitations of ML methods you stated do present challenges to some applications, particularly in high-stakes scenarios. We hope to incorporate some of these consideration in future work, and deeply appreciate you taking the time to provide feedback!

---

### Official Review · Reviewer_cQ8K · 2024-11-04

**Soundness:** 3
**Presentation:** 4
**Contribution:** 4
**Rating:** 8
**Confidence:** 3

**Summary:**

The paper presents an alternative formulation for a trainable network based on the KST.

The paper presents the theoretical property of the model and experimentally validates the performance as an approximation function of PINN, on 3 PDEs.

**Strengths:**

The paper is clear and justify its contribution, and put in context with the current state of art

The paper justifies the change in the KAN architecture and its relationship. The connection with the multi-head transformer is a bit stretched tho.

The paper provides some experiments to show the potential of changing the representation of the KST.

**Weaknesses:**

The exposition is very good, the experiments show the advantage with respect to other KST architecture.

There is only the evaluation in the PINN context. It would be nice to have more experiments, for example against some neural operators and training from data.

**Questions:**

I would like to see, even in the annex, the behavior against Neural Operators.

is there other choices of the b(t) functions that works well?

How do you choose the hyper-parameters (N,m)?

**Details Of Ethics Concerns:**

No ethical concerns were foreseen.

---

> ### Author Response · Authors · 2024-11-21
> **Author Response**
>
> First of all, we thank you for your careful review and comments! We deeply appreciate your positive review and hope to address your main points below:
> - While we agree that evaluating ActNet in data-driven problems and neural operator learning is an important research direction, we decided to focus this work on establishing ActNet's capabilities in the physics-informed setting first. We are actively investigating principled ways to extend ActNet to neural operator architectures by designing appropriate conditioning and modulation mechanisms that maintain the theoretical foundations from Kolmogorov theory. Rather than simply replacing MLPs with ActNet blocks in existing architectures, we believe a more systematic approach is needed to fully leverage ActNet's unique properties in the operator learning setting. This represents an exciting direction for future work that could potentially combine the strengths of both frameworks.
> - During initial exploratory phases of our work, we also explored using polynomial basis functions in the implementation of ActNet. Overall, this approach also worked, but slightly less robustly than using the sinusoidal basis. For the sake of simplicity, we decided to stick to a single type of basis function, but as future work we may include other basis possibilities as well.
> - We have added a section to the appendix detailing the results of some hyperparameter ablations. In short, we find that the basis size $N$ does not need to be large for accurate results. In fact, except for small parameter counts, we don’t see accuracy gains for increasing the value of $N$ beyond 4 or so, while the computational complexity of the model increases for larger $N$ (even when network size is fixed). Additionally, as is often the case for deep learning, we find that composing more layers and increasing network width generally improves performance by increasing network capacity.

---

> > ### Comment · Reviewer_cQ8K · 2024-11-26
> > **thank you for your comments**
> >
> > Dear Authors,
> >
> > I thank you for your reply.
> >
> > I keep my positive evaluation.
> >
> > Kind Regards

---

### Author Response · Authors · 2024-11-21
**Summary of Improvements**

## **General Post To Reviewers**
First of all, we would like to thank our reviewers and area chairs for their thoughtful feedback and insights. Based on the reviewers' comments, we have made several significant improvements to our paper, summarized as follows. The new version of the PDF file includes changes highlighted in blue text.

### **Additional Experiments and Comparisons**
**Expanded Comparisons with State-of-the-Art**: We now include comprehensive comparisons against PirateNet, a leading PINN architecture with sophisticated inductive biases. Our results show ActNet outperforms PirateNet on both the Advection equation and full-time solution of the Kuramoto-Sivashinsky equation. Notably, ActNet achieves this superior performance without using PirateNet's physics-informed initialization and adaptive residual connections (features that could potentially further improve ActNet's performance in future work).

**Broader MLP Baselines**: We have added traditional MLPs with tanh, sigmoid and GELU activations as additional baselines. After running over 1,700 new experiments across different architectures and hyperparameter settings, we find ActNet consistently outperforms these baselines, particularly on challenging problems with high-frequency content. In total, our evaluation now encompasses over 7,000 independent training runs, demonstrating a rigorous empirical validation of our approach.

### **New Hyperparameter Analysis**
We have added a detailed analysis of ActNet's hyperparameter choices in Appendix D.4, providing practical guidance for implementing our method. Key findings include:
- The basis size N=4 is typically sufficient, with limited gains from larger values.
- Network width and depth trade-offs follow expected deep learning patterns.
- Minimum of two ActLayers is recommended, aligning with our theoretical analysis.

### **Enhanced Theoretical Clarity**
We have improved our theoretical exposition by:
- Correcting notation and typos in proofs.
- Adding explanatory details to make derivations more accessible.
- Clarifying connections between theoretical guarantees and practical implementation.

We also plan to make all experimental data, including loss curves and training runs, publicly available through WeightsAndBiases (currently withheld for anonymity).

---

### Author Response · Authors · 2024-11-25
**Reminder About Discussion Period Deadline**

First of all, we would like to thank the reviewers for their time. We would also like to remind you that last week we published improvements to our paper (including additional experiments and ablations) and responded to the questions from your reviews. Given that **tomorrow, November 26th**, is the last day for reviewers to post comments, we would like to remind you of this deadline, and encourage you to ask any more clarifying questions you might have. So far, only one of the reviewers has responded to our replies, and improved their score significantly, which we believe is indicative of the significance of our new additions to the paper.

---

### Author Response · Authors · 2024-12-03
**Summary of Paper And Discussion Period**

# Summary of Paper And Discussion Period

## Paper Summary
Our  paper explores alternative formulations of the Kolmogorov Superposition Theorem (KST) for deep learning applications and introduces **ActNet**, a scalable neural network architecture that addresses many of the practical limitations of KANs and other KST-inspired methods. Evaluated in the context of Physics-Informed Neural Networks (PINNs) for solving partial differential equations, a well-suited testbed for approaches based on the KST, ActNet consistently outperforms KANs and competes with state-of-the-art MLP-based models such as PirateNets, highlighting its potential for scientific computing and PDE simulations. Overall, we believe our paper successfully proposes an interesting and novel theoretical perspective on applications of the KST to deep learning, and supports this new direction with several rigorous experiments and ablations that showcase ActNet’s promise in physics-informed machine learning. We hope this work leads other researchers to use ActNet in challenging PINN tasks, and to consider their own alternatives to our method, by potentially adapting further formulations of the KST.

## Discussion Period Summary
We would like to begin by thanking all the reviewers and area chairs for facilitating the discussion period and helping us strengthen our paper. After receiving initial reviews we took the time to carefully consider each of the reviewer's questions and concerns, and based on this feedback we carried out new experiments and provided additional information in the manuscript. We believe this lead to a considerable improvement to our submission by helping highlight the advantages of our method over existing architectures, in addition to providing better theoretical clarity.

In the rebuttal version of our paper we:
- **Expanded comparisons against state-of-the-art architectures** by including results for PirateNet, which currently offer some of the best performance for PINNs in challenging settings. We find that a plain ActNet achieves superior performance without even using PirateNet's physics-informed initialization and adaptive residual connections (features that could potentially further improve ActNet's performance in future work).
- **Included broader MLP baselines** in ablation experiments by running MLPs with tanh, sigmoid and GELU activations as additional baselines. After running over 1,700 new experiments across different architectures and hyperparameter settings, we find ActNet consistently outperforms these baselines, particularly on challenging problems with high-frequency content. In total, our evaluation now encompasses over 7,000 independent training runs, demonstrating a rigorous empirical validation of our approach, in addition to the strong theoretical component of our work.
- **Added a new appendix analysing hyperparameter choice**, which clarifies the influence of architecture hyperparameters on performance. Notably, we find that the basis size $N$ does not need to be large in order to yield good results (with $N=4$ typically sufficing), and that there is significant improvement when using at least two ActLayers, a phenomena which is directly related to the theoretical guarantees provided by the Laczkovich version of the KST as described in theorem 3.3.
- **Improved the exposition of theoretical elements** by introducing new explanations to steps in proofs and correcting typos and notation mistakes.

We believe these improvements were possible due in no small part to the quality of the reviews, and we deeply thank reviewers and area chairs for taking the time to make the discussion period productive. Reviewers appeared to be supportive of the changes we included in our rebuttal, **with the average reviewer score increasing by 1.5 points**, which we believe is indicative of the hard work we put into the submission, as well as the quality of our paper.

---

### Meta-Review · Area_Chair_e64G · 2024-12-22

**Metareview:**

Based on a variant of Kolmogorov superposition theorem given by Miklos Laczkovich in 2021, the authors propose to use it as a deep neural network design. It used d inner functions, and one outer function. The ActNet based on this has d adaptive activation functions, different for each neuron. The numerical experiments are quite sparse in this case: 5 problems vs 4 methods However, the experiments themselves seem to be done correctly, for different number of parameters/seeds.
Overall, although:
1) No theoretical results (i.e. are the 'true' functions smooth, so they can be approximated well with simple basis functions?)
2) Numerical experiments are not 100% convincing that this approach is better

all the reviewers are quite positive in the end.

**Additional Comments On Reviewer Discussion:**

The rebuttal really changed the initial grades (+1.5 points), so it went well. I am not completely thrilled by the results (i.e. pick a theorem and rewrite it as a network, but the crucial problem of KST is still there - are those functions in the exact representations really smooth to be approximated, for example, in the Fourier basis?), but can not go against the very high grades of the reviewers.

---

### Decision · Program_Chairs · 2025-01-22

Accept (Spotlight)